# Boson-fermion pairing and condensation in two-dimensional Bose-Fermi mixtures

**Leonardo Pisani[1,2]\*, Pietro Bovini[1,2], Fabrizio Pavan[3] and Pierbiagio Pieri[1,2]†**

**1** INFN, Sezione di Bologna, Viale Berti Pichat 6/2, I-40127, Bologna, Italy
**2** Dipartimento di Fisica e Astronomia "Augusto Righi", Università di Bologna,
Via Irnerio 46, I-40126, Bologna, Italy
**3** Dipartimento di Fisica E. Pancini - Università di Napoli Federico II - I-80126 Napoli, Italy

⋆ leonardo.pisani2@unibo.it , † pierbiagio.pieri@unibo.it

## Abstract

We consider a mixture of bosons and spin-polarized fermions in two dimensions at zero temperature with a tunable Bose-Fermi attraction. By adopting a diagrammatic $T$-matrix approach, we analyze the behavior of several thermodynamic quantities for the two species as a function of the density ratio and coupling strength, including the chemical potentials, the momentum distribution functions, the boson condensate density, and the Tan's contact parameter. By increasing the Bose-Fermi attraction, we find that the condensate is progressively depleted and Bose-Fermi pairs form, with a small fraction of condensed bosons surviving even for strong Bose-Fermi attraction. This small condensate proves sufficient to hybridize molecular and atomic states, producing quasi-particles with unusual Fermi liquid features. A nearly universal behavior of the condensate fraction, the bosonic momentum distribution, and Tan's contact parameter with respect to the density ratio is also found.

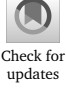

# 1 Introduction

Mixed systems of bosons and fermions are at the base of our understanding of the physical world. They feature in the Standard Model of elementary-particle physics as gauge and matter fields, respectively, in high-density quark matter as diquarks interacting with unpaired quarks [1], in nuclear matter in the interacting boson-fermion model for atomic nuclei with an odd number of protons or neutrons [2], and as boson-mediated interactions in soft [3] as well as condensed [4] matter. Ultra-cold atomic gases provide an exceptionally versatile platform for the simulation of quantum matter [5]. In the case of Bose-Fermi (BF) atomic mixtures, several realizations [6–39] have allowed exploration of a wide range of phenomena like phase separation [22], polarons [13, 17, 23, 26, 31], dual superfluidity [15, 16, 19, 20], collective excitations [34, 39], mediated interactions [28, 35–37], and Feshbach molecules [6, 8, 10, 25, 30, 33].

In parallel, BF mixtures have also been realized in semiconductor nanostructures due to the recent advent of atomically thin transition metal dichalcogenides (TMD) [40–45]. These materials intrinsically offer a remarkable opportunity to investigate the physics in two dimensions (2D) where topological phases play a crucial role [46, 47]. The so-called spin-valley locking, originating from the strong spin-orbit coupling of the transition metal atoms, allows for a complete distinguishability between electrons (and/or holes) populating the two spin-polarized valleys at Brillouin points $K$ and $K'$ [48]. Exciton formation and charge doping are therefore two distinct processes that affect separate assemblies of electrons (and/or holes). In this scenario, the production of a BF mixture in a two-dimensional solid-state setting becomes therefore possible, with bosons corresponding to tightly bound excitons and fermions to doped charges. The recent detection of electrically tunable two-dimensional Feshbach resonances in TMD bilayers has put this target on more solid ground [43, 49].

However, the unique flexibility of atomic gases is unrivaled when the need arises to manipulate dimensionality and interaction strength at the same time. Confinement-induced resonances accomplish this purpose by allowing for the realization of a dimensional crossover, starting from 3D confinement down to (quasi-)2D one. If the energy level spacing of the 3D trapping system is kept much larger than the energy scale of the intervening scattering processes, the system becomes effectively 2D and the ensuing 2D scattering length can be manipulated by varying either the scattering length associated with the original 3D Feshbach resonance and/or the confinement length, yet ensuring that the quasi-2D condition above remains satisfied [50–52]. This situation is particularly apt for BF mixtures, for which one would need to adjust independently the BF and the boson-boson (BB) interactions owing to the mechanical stability of the assembly being conditioned primarily by that of the Bose component [53]. Having at one's disposal two independent degrees of freedom in the 3D scattering length and the confinement length, one can then correspondingly tune the effective BF and BB interactions (even though in a highly non-linear fashion [50]) and simultaneously analyze the stability of different phases of the mixture across the whole resonant regime.

Under such circumstances, a recent work has shown that two-dimensional weakly bound fermionic molecules formed in a BF mixture with attractive BF and repulsive BB interactions are expected to be collisionally stable and should exhibit a strong $p$-wave mutual attraction [54]. In fact, three-body recombination, which has historically plagued these mixed systems in three dimensions, is concomitantly avoided because Efimovian processes are suppressed in 2D.

$P$-wave superfluidity is intensely pursued in both atomic-based [55] and semiconductor-based [46,56] mixtures, as it opens the way to the exploration of topologically protected states, including Majorana modes and non-Abelian vortex excitations [57,58], hence offering a route for the implementation of fault-tolerant quantum computation [59–64].

Within the above scenario, it is crucial to explore BF mixtures with a tunable BF attraction in a 2D setting. Until now, theoretical efforts have concentrated mainly on 3D settings [65–86]. In 2D, theoretical work has been sparse so far [87–90] and has primarily dealt with phase competition in optical lattices [87,89], phase stability in atomic traps [88], or with the bosonic impurity limit of a boson density $n_B$ much smaller than the fermion density $n_F$ [90].

In this work, we study a 2D BF mixture in homogeneous space with boson and fermion densities spanning from the bosonic impurity limit ($n_B \ll n_F$) to the matched density regime ($n_B = n_F$). Similarly to the analyses carried out in 3D [76, 80, 83, 84, 86, 91], we are here interested in the competition between boson condensation and BF pairing in mixtures with an attractive and tunable BF interaction at zero temperature. For this reason, we consider concentrations of bosons $x = n_B/n_F \leq 1$, such that a full competition between pairing and condensation is allowed. The BB interaction (when considered) is taken as repulsive for mechanical stability reasons. Numerical calculations focus on the case of equal boson and fermion masses in order to reduce the computational effort. However, to facilitate possible future studies, the theoretical formalism is developed for generic boson and fermion masses $m_B$ and $m_F$.

Our main results are as follows. (i) We find that the condensate is progressively depleted as the BF attraction increases, in analogy to the 3D case. However, while in 3D the condensate vanishes beyond a critical coupling strength, in 2D the condensate does not exactly vanish but rather becomes exponentially small with the coupling strength. (ii) Similarly to the 3D case, we uncover a nearly universal behavior of the condensate fraction, the bosonic momentum distribution, and Tan's contact parameter with respect to the boson concentration. (iii) BF pairs form for sufficiently strong BF attraction. Quite generally, these composite fermions are hybridized states between a pure molecular bound state and an unpaired state made of a fermionic atom and a boson belonging to the condensate. We find that these hybridized states produce quasi-particles with rather peculiar Fermi liquid features. (iv) The boson momentum

distribution function shows quite a rich behavior when the BF coupling strength and boson concentration are varied, which could in principle be tested in future experiments with BF mixtures.

The article is organized as follows. In Sec. 2 we introduce the model Hamiltonian, illustrate the diagrammatic theory of choice and lay out the full set of thermodynamic equations to solve. In Sec. 3 we explore the strong-coupling limit for the BF coupling parameter. In this limit, a set of semi-analytical and rather simple equations can be derived, allowing us to obtain crucial insights into the microscopic physics and phenomenology of a strongly attractive BF mixture in 2D. In Sec. 4 we present our numerical results for the boson and fermion momentum distributions, corresponding chemical potentials, condensate density and Tan's contact parameter. In Sec. 5 we discuss the virtues and limitations of our approach and outline perspectives for future work.

## 2 Formalism

### 2.1 The system Hamiltonian

We consider a mixture of bosons (B) and single-component fermions (F) with interspecies attraction in a two-dimensional homogeneous setting and focus on the zero temperature limit in which bosons may condense even in 2D [92].

The system is assumed to be dilute such that the range of all interactions is smaller than the average inter-particle distance. Most common regimes of current experiments on BF mixtures involve the presence of a broad Fano-Feshbach resonance, whereby the boson-fermion scattering length is larger than the interaction range. This justifies the adoption of an effective pseudo-potential of the point-contact form [93] as far as the modeling of the BF interaction is concerned. The BB interaction is assumed to be short-ranged and weakly repulsive in order to make the system mechanically stable. Hence, within a minimal model approach, the resulting Hamiltonian in the grand-canonical ensemble has the form

$$H = \sum_{s=\mathrm{B,F}} \int d\mathbf{r} \, \psi_s^\dagger(\mathbf{r}) \left( -\frac{\nabla^2}{2m_s} - \mu_s \right) \psi_s(\mathbf{r}) + v_0^{\mathrm{BF}} \int d\mathbf{r} \, \psi_\mathrm{B}^\dagger(\mathbf{r}) \psi_\mathrm{F}^\dagger(\mathbf{r}) \psi_\mathrm{F}(\mathbf{r}) \psi_\mathrm{B}(\mathbf{r})$$
$$+ \frac{1}{2} \int d\mathbf{r} \int d\mathbf{r}' \, \psi_\mathrm{B}^\dagger(\mathbf{r}) \psi_\mathrm{B}^\dagger(\mathbf{r}') U_{\mathrm{BB}}(\mathbf{r}-\mathbf{r}') \psi_\mathrm{B}(\mathbf{r}') \psi_\mathrm{B}(\mathbf{r}), \tag{1}$$

where $\psi_s^\dagger$ and $\psi_s$ respectively create and destroy a particle of mass $m_s$ and chemical potential $\mu_s$, with $s = \mathrm{B,F}$ indicating the bosonic or fermionic nature. We set $\hbar = 1$ throughout this paper.

The first term in Eq. (1) describes a system of non-interacting single-component bosons and fermions. The second term accounts for the boson-fermion interaction, which is attractive with coupling strength $v_0^{\mathrm{BF}}$. The third term deals with the boson-boson interaction, which is taken to be weakly repulsive for stability reasons and can be treated perturbatively at the level of the Bogoliubov approximation for the two-dimensional Bose gas (as originally developed by Schick [94]). This boils down to adopting the effective replacement $U(\mathbf{r}-\mathbf{r}') \to v_0^{\mathrm{BB}} \delta(\mathbf{r}-\mathbf{r}')$ with an effective BB coupling strength $v_0^{\mathrm{BB}} = 4\pi\eta_\mathrm{B}/m_\mathrm{B}$, where $\eta_\mathrm{B} = -1/\ln(n_\mathrm{B} a_{\mathrm{BB}}^2)$ is the (small) gas parameter and $a_{\mathrm{BB}}$ is the (2D) BB scattering length. Finally, the $s$-wave fermion-fermion scattering is forbidden by the Pauli exclusion principle ($v_0^{\mathrm{FF}} = 0$).

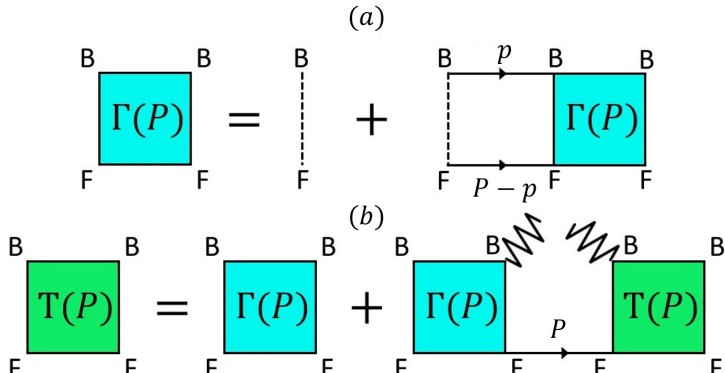

Figure 1: Feynman's diagrams for (a) the particle-particle ladder $\Gamma(P)$ and (b) the $T$-matrix in the condensed phase $T(P)$. Full lines correspond to bare boson (B) and fermion (F) Green's functions, dashed lines to bare BF interactions and zigzag lines to condensate factors $\sqrt{n_0}$.

The BF attractive contact interaction needs to be regularized by expressing the bare strength $v_0^{\mathrm{BF}}$ in terms of the scattering length $a_{\mathrm{BF}}$. This is achieved by the equation

$$\frac{1}{v_0^{\mathrm{BF}}} = -\int \frac{d\mathbf{k}}{(2\pi)^2} \frac{1}{\varepsilon_0 + \frac{k^2}{2m_r}} \, , \tag{2}$$

relating the bare coupling strength $v_0^{\mathrm{BF}}$ with the boson-fermion binding energy $\varepsilon_0 = 1/(2m_r a_{\mathrm{BF}}^2)$ of the associated two-body problem in vacuum, where $a_{\mathrm{BF}}$ is the (2D) BF scattering length and $m_r = m_{\mathrm{B}} m_{\mathrm{F}}/(m_{\mathrm{B}} + m_{\mathrm{F}})$ is the reduced mass of the boson-fermion system. The ultraviolet-divergent integral on the right-hand side of Eq. (2) compensates analogous divergences occurring in many-body diagrammatic perturbation theory (see, e.g., [95] and Supplemental Material of [96] for a discussion of the regularization procedure).

In analogy with 2D fermionic systems, we introduce an effective boson-fermion coupling parameter $g = -\ln(k_{\mathrm{F}} a_{\mathrm{BF}})$, where $k_{\mathrm{F}} = \sqrt{4\pi n_{\mathrm{F}}}$ is the 2D Fermi momentum of a non-interacting Fermi gas with the same density $n_{\mathrm{F}}$ of the fermionic component. The weak and strong BF coupling regimes are defined by $g \ll -1$ and $g \gg 1$ but, in practice, they are well represented by the effective ranges, $g \lesssim -2$ and $g \gtrsim 2$, respectively.

On physical grounds, the following picture is expected when the BF coupling strength is varied. In the weak-coupling limit, bosons are expected to be almost fully condensed, with a small depletion of the condensate density $n_0$ determined just by the weak BB repulsion [94] for $g \to -\infty$. Fermions instead fill up a Fermi sphere. In the strong-coupling limit, bosons are instead expected to be expelled out of the condensate and form molecular fermions that fill up a Fermi sphere. We will discuss below to what extent, within the present approach, this simple picture continues to hold also in two dimensions.

## 2.2 Many-body $T$-matrix

The diagrammatic many-body approach adopted in the present work relies on previous works on BF mixtures in 3D, both above [76] and below [91] the BEC transition temperature, based on the non-self-consistent $T$-matrix (ladder) approximation. In these works, such approximation was shown to recover the well-known polaron-to-molecule transition of an impurity in a polarized Fermi gas as the coupling strength is increased, and to predict a universal behavior of the condensate fraction with respect to the boson concentration (from $x \to 0$ to $x = 1$) as the coupling strength is varied. Such outcomes were recently confirmed in a breakthrough

$(a)$ $\qquad\qquad\qquad$ $(b)$

$$\Sigma_{BF}(q) = \text{[diagram]} \qquad \Sigma_F(k) = \text{[diagram]} + \text{[diagram]}$$

Figure 2: Feynman's diagrams for the boson self-energy $\Sigma_{\text{BF}}$ arising from interactions with fermions (a) and fermion self-energy $\Sigma_{\text{F}}$ (b). Full lines correspond to bare boson and fermion Green's functions, and zigzag lines to condensate factors $\sqrt{n_0}$.

experiment on 3D BF mixtures, where for the first time the regime of double degenerate BF mixtures with matching densities was reached [33]. There, the authors observed the existence of a quantum phase transition between a polaronic condensed phase and a molecular phase and, even for nearly matched densities, confirmed the predictions of Ref. [91] on the universal character of the condensate fraction with respect to the boson concentration. These successes in 3D, motivate us to keep the same diagrammatic choice of [91] and extend it to the 2D case.

A warning is however in order. In the limiting case of a single impurity in a Fermi gas, the present approach it has been shown to miss the polaron-to-molecule transition in 2D [97], which is found instead by more sophisticated approaches [98, 99]. We should thus bear in mind that such a shortcoming of the present approximation, which is due to an overestimate of the atom-molecule repulsion in the strong-coupling limit of the theory, may also extend at finite boson concentration $x$, in particular in the limit $x \to 0$ that approaches the single impurity case.

Let us now see the rationale behind the choice of diagrams in [91], which we wish to extend to 2D. Since the pioneering work by Nozières and Schmitt-Rink [100] on the related problem of the BCS-BEC crossover in two-component Fermi gases, it is well known that ladder diagrams are able to capture pairing (molecular) correlations in the normal phase (see also [101] for a review). Only after the inclusion of this class of diagrams does the superfluid critical temperature recover the Bose-Einstein condensation temperature in the strong-coupling limit of the BCS-BEC crossover. In addition, for the same contact potential we are considering, ladder diagrams provide the leading self-energy in the weak-coupling limit of the BCS-BEC crossover [102]. They thus provide a sensible scheme to describe the whole BCS-BEC crossover, even in the intermediate coupling region in which fully controlled approximations are not available.

The same strategy is then adopted for the present problem, in which we are interested in setting up a theory able to describe the progressive formation of paring (molecular) correlations in a Bose-Fermi mixture when the BF interaction is varied from weak to strong. When switching to this problem, the required modification is straightforward in the normal phase: the particle-particle ladder made of the repeated interaction of spin-up and spin-down fermions is replaced by a particle-particle ladder made of the repeated interaction of bosons with (one-component) fermions [76]. This corresponds to the particle-particle ladder $\Gamma(P)$ in Fig. 1(a), where continuous lines indicate bare fermionic or bosonic propagators, while dashed lines indicate the bare boson-fermion interaction $v_0^{\text{BF}}$. In the zero temperature limit, Feynman rules yield

$$\Gamma(\mathbf{P},\Omega) = v_0^{\text{BF}} - v_0^{\text{BF}}\Gamma(\mathbf{P},\Omega) \int \frac{d\mathbf{p}}{(2\pi)^2} \int_{-\infty}^{\infty} \frac{d\omega}{2\pi} G_{\text{F}}^0(\mathbf{P}-\mathbf{p},\Omega-\omega) G_{\text{B}}^0(\mathbf{p},\omega), \tag{3}$$

where $G_B^0$ and $G_F^0$ are bare Green's functions for bosons and fermions

$$G_s^0(\mathbf{k},\omega) = \frac{1}{i\omega - \xi_{\mathbf{k}}^s}, \qquad s = \mathrm{B}, \mathrm{F}, \tag{4}$$

with $\xi_{\mathbf{k}}^s = k^2/2m_s - \mu_s$.

Integrating over $\omega$ and solving for $\Gamma(\mathbf{P},\Omega)$ in Eq. (3), one obtains

$$\Gamma(\mathbf{P},\Omega)^{-1} = \frac{1}{v_0^{\mathrm{BF}}} + \int \frac{d\mathbf{k}}{(2\pi)^2} \frac{1 - \Theta\left(-\xi_{\mathbf{P-k}}^{\mathrm{F}}\right) - \Theta\left(-\xi_{\mathbf{k}}^{\mathrm{B}}\right)}{\xi_{\mathbf{P-k}}^{\mathrm{F}} + \xi_{\mathbf{k}}^{\mathrm{B}} - i\Omega}, \tag{5}$$

whereby the $\Theta$ functions appearing in the integrand correspond to the Fermi and Bose distributions in the zero temperature limit. In particular, for the concentrations $x \le 1$ and the range of BF and BB coupling strengths considered in this work, $\mu_{\mathrm{B}}$ is always negative, and hence $\Theta(-\xi_{\mathbf{k}}^{\mathrm{B}}) = 0$.

One then obtains (see Appendix A for details)

$$\Gamma(\mathbf{P},\Omega) = \frac{1}{T_2(\mathbf{P},\Omega)^{-1} - I_{\mathrm{F}}(\mathbf{P},\Omega)}, \tag{6}$$

where $T_2$ is the off-shell two-body $T$-matrix in vacuum and the contribution $I_{\mathrm{F}}(\mathbf{P},\Omega)$ stems from the presence of the fermionic medium.

Note that the present diagrammatic approach corresponds to the zero-temperature limit of the Matsubara formalism at finite temperature. We will thus work on the imaginary frequency axis rather than on the real frequency one (as instead it is standard practice at $T = 0$ [103]). This choice is particularly useful to avoid singularities of the Green's function on the real frequency axis.

We now consider the condensed phase. In this case, one has to take into account the possibility that fermions repeatedly interact also with condensed bosons, besides non-condensed bosons. By summing all possible combinations of the repeated scattering of fermions with condensed or non-condensed bosons one obtains the many-body $T$-matrix in the condensed phase described by the Feynman diagram of Fig. 1(b). The resulting Bethe-Salpeter equation for the many-body $T$-matrix in the condensed phase $T(\mathbf{P},\Omega)$ thus reads

$$T(\mathbf{P},\Omega) = \Gamma(\mathbf{P},\Omega) + n_0 \Gamma(\mathbf{P},\Omega) G_F^0(\mathbf{P},\Omega) T(\mathbf{P},\Omega), \tag{7}$$

which immediately yields

$$T(\mathbf{P},\Omega) = \frac{1}{\Gamma(\mathbf{P},\Omega)^{-1} - n_0 G_F^0(\mathbf{P},\Omega)}. \tag{8}$$

## 2.3 Boson and fermion self-energies

We now discuss the bosonic and fermionic self-energies within the present $T$-matrix approach.

The bosonic self-energy is made of two terms: originating, respectively, from BF and BB interactions. For the former, we proceed as in the corresponding problem in the BCS-BEC crossover: the self-energy of one species is obtained by closing the $T$-matrix by a propagator of the other species.

This is shown in Fig. 2(a), yielding the expression

$$\Sigma_{\mathrm{BF}}(\mathbf{k},\omega) = \int \frac{d\mathbf{P}}{(2\pi)^2} \int \frac{d\Omega}{2\pi} T(\mathbf{P},\Omega) G_F^0(\mathbf{P-k},\Omega-\omega) e^{i\Omega 0^+}, \tag{9}$$

where the factor $e^{i\Omega 0^+}$ resulting from Feynman's rules [103] ensures convergence of the frequency integral. Note here that we adopt a sign convention such that the many-body $T$-matrix $T(\mathbf{P},\Omega)$ reduces to the two-body $T$-matrix in vacuum [104].

To ensure the mechanical stability of the bosonic component of the mixture, a minimal repulsive interaction between bosons is needed. This repulsion is described within the standard Bogoliubov approximation for weakly interacting bosons [105]. Extension of the latter to two dimensions was originally done by Schick in [94] (and later by Popov [106]), who showed that the lowest order anomalous and normal self-energies have the form

$$\Sigma_{12} = \frac{-4\pi n_0}{m_{\mathrm{B}} \ln\left(n_{\mathrm{B}} a_{\mathrm{BB}}^2\right)}, \qquad \Sigma_{11} = 2\Sigma_{12}, \tag{10}$$

with the natural definition of the small gas parameter $\eta_{\mathrm{B}} = -1/\ln\left(n_{\mathrm{B}} a_{\mathrm{BB}}^2\right)$ which identifies the boson-boson interaction strength.

In principle, a Bogoliubov treatment of the boson-boson interaction, which implicitly assumes a large condensate fraction, may appear questionable for strong boson-fermion couplings, for which the condensate fraction is significantly depleted. In this regime, however, stability is expected to be guaranteed by fermionization of the BF mixture into a mixture of composite and unpaired fermions, thus making the contribution of $\eta_{\mathrm{B}}$ in practice immaterial.

The resulting expression of the normal boson self-energy acquires the final form

$$\Sigma_{\mathrm{B}}(\mathbf{k},\omega) = \Sigma_{11} + \Sigma_{\mathrm{BF}}(\mathbf{k},\omega). \tag{11}$$

For the fermionic self-energy in the condensed phase, the bosonic line closing the many-body $T$-matrix $T(\mathbf{P},\Omega)$ can be either a condensed line or a non-condensed one. In the first case, however, the many-body $T$-matrix in the condensed phase $T(\mathbf{P},\Omega)$ needs to be replaced by $\Gamma(\mathbf{P},\Omega)$ in order for the self-energy to be irreducible (and thus avoiding a double-counting of diagrams when inserting the self-energy in the Dyson equation). The fermionic self-energy is thus described by the Feynman diagrams of Fig. 2(b), corresponding to the expression

$$\Sigma_{\mathrm{F}}(\mathbf{k},\omega) = n_0 \Gamma(\mathbf{k},\omega) - \int \frac{d\mathbf{P}}{(2\pi)^2} \int \frac{d\Omega}{2\pi} T(\mathbf{P},\Omega) G_{\mathrm{B}}^0(\mathbf{P}-\mathbf{k},\Omega-\omega) e^{i\Omega 0^+}. \tag{12}$$

## 2.4 Green's functions, momentum distributions, and densities

Once the bosonic and fermionic self-energies are identified, Dyson's equations for the corresponding Green's functions yield

$$G_{\mathrm{F}}^{-1}(\mathbf{k},\omega) = G_{\mathrm{F}}^0(\mathbf{k},\omega)^{-1} - \Sigma_{\mathrm{F}}(\mathbf{k},\omega), \tag{13}$$

$$G_{\mathrm{B}}^{-1}(\mathbf{k},\omega) = i\omega - \xi_{\mathbf{k}}^{\mathrm{B}} - \Sigma_{\mathrm{B}}(\mathbf{k},\omega) + \frac{\Sigma_{12}^2}{i\omega + \xi_{\mathbf{k}}^{\mathrm{B}} + \Sigma_{\mathrm{B}}(-\mathbf{k},-\omega)}. \tag{14}$$

The momentum distributions for fermions and out-of-condensate bosons are obtained in the standard way

$$n_{\mathrm{F}}(\mathbf{k}) = \int_{-\infty}^{+\infty} \frac{d\omega}{2\pi} G_{\mathrm{F}}(\mathbf{k},\omega) e^{i\omega 0^+}, \tag{15}$$

$$n_{\mathrm{B}}(\mathbf{k}) = -\int_{-\infty}^{+\infty} \frac{d\omega}{2\pi} G_{\mathrm{B}}(\mathbf{k},\omega) e^{i\omega 0^+}, \tag{16}$$

and so their respective number densities

$$n_{\text{F}} = \int \frac{d\mathbf{k}}{(2\pi)^2} n_{\text{F}}(\mathbf{k}), \tag{17}$$

$$n_{\text{B}} = n_{\text{B}}' + n_0 = \int \frac{d\mathbf{k}}{(2\pi)^2} n_{\text{B}}(\mathbf{k}) + n_0. \tag{18}$$

Finally, due to the presence of a bosonic component at zero temperature, the Hugenholtz-Pines condition ( [107–109]) for the corresponding chemical potential is imposed as

$$\mu_{\text{B}} = \Sigma_{\text{B}}(\mathbf{k} = \mathbf{0}, \omega = 0) - \Sigma_{12}. \tag{19}$$

The three equations (17)-(19) are at the core of our study. They constitute a system of non-linear integral equations whereby the condensate density and the bosonic and fermionic chemical potentials are solved for, once the values of the densities $n_{\text{B}}, n_{\text{F}}$ and scattering lengths $a_{\text{BF}}$ and $a_{\text{BB}}$ are given.

## 3 Strong boson-fermion coupling regime

In this section, we will examine the regime of strong boson-fermion coupling $\varepsilon_0/E_{\text{F}} \equiv 2e^{2g} \gg 1$, where a simplified and more transparent approach can be adopted. The system of Eqs. (17)-(19) is replaced with an equivalent but much simpler one, for which a semi-analytical solution can be found. This allows us to disclose the microscopic physics underlying the different concentration regimes ($0 \leq x \leq 1$) in the strong-coupling regime ($g > 2$), and, at the same time, to perform stringent checks on crucial physical quantities otherwise obtained through a full numerical implementation. The direct boson-boson repulsion is expected to produce only minor effects in this regime, which is dominated by BF pairing and the corresponding large binding energy. For this reason, we altogether neglect it in the present section.

### 3.1 Many-body $T$-matrix in the condensed phase $T(\mathbf{P}, \Omega)$

We first delve into the nature of the molecular states by analyzing the many-body $T$-matrix in the condensed phase $T(\mathbf{P}, \Omega)$ which, as we will see, for strong boson-fermion coupling, acquires the meaning of a composite fermion propagator.

In the limit $\varepsilon_0/E_{\text{F}} \gg 1$, the two contributions in Eq. (6) to $\Gamma(\mathbf{P}, \Omega)$ acquire a simplified form. The contribution $T_2(\mathbf{P}, \Omega)$ reduces to the polar form (see Appendix A, Eq. (A.11))

$$T_2(\mathbf{P}, \Omega) \approx \frac{2\pi\varepsilon_0}{m_r} \frac{1}{i\Omega - \frac{P^2}{2M} + \mu_{\text{CF}}}, \tag{20}$$

where $\mu_{\text{CF}} \equiv \mu_{\text{F}} + \mu_{\text{B}} + \varepsilon_0$ and $M \equiv m_{\text{B}} + m_{\text{F}}$ correspond to the chemical potential and mass of the composite fermions, respectively.

The contribution $I_{\text{F}}(\mathbf{P}, \Omega)$ can instead be approximated as (see Appendix A, Eq. (A.13))

$$I_{\text{F}}(\mathbf{P}, \Omega) \approx \frac{n_{\mu_{\text{F}}}^0}{\varepsilon_0}, \tag{21}$$

where $n_{\mu_{\text{F}}}^0 = m_{\text{F}} \mu_{\text{F}} \Theta(\mu_{\text{F}})/2\pi$ is the number density of a non-interacting Fermi gas with chemical potential $\mu_{\text{F}}$.

When Eqs. (20) and (21) are inserted into Eq. (6), the latter acquires the strong-coupling form

$$\Gamma_{\text{SC}}(\mathbf{P},\Omega) \approx \frac{2\pi\varepsilon_0}{m_r} \frac{1}{i\Omega - \frac{P^2}{2M} + \mu_{\text{CF}} - \frac{2\pi}{m_r}n_{\mu_F}^0} \,, \tag{22}$$

which suggests the presence of an effective self-energy term $\Sigma_{\text{CF}}^0 = 2\pi n_{\mu_F}^0 / m_r$. The latter represents an effective repulsion field generated by the medium of atomic fermions ($\mu_F > 0$) onto the existing molecules and is an effective way to take into account the kinetic energy cost of populating the molecular Fermi sphere in addition to the atomic one.

In 2D the three-body $T$-matrix in vacuum, when treated within the Born approximation, reduces to a constant value $2\pi/m_r$, implying that $\Sigma_{\text{CF}}^0$ is a Hartree contribution of the interaction between the molecular and atomic species (see Appendix B). The atom-molecule scattering length is actually expected to vanish in the strong-coupling limit of the BF interaction [110]. The coupling-independent atom-molecule scattering length obtained by the present approach thus overestimates the atom-molecule repulsion.

Being a mere mean-field shift, $\Sigma_{\text{CF}}^0$ can be absorbed into a redefinition of the chemical potential in Eq. (20), $\mu_{\text{CF}} \rightarrow \tilde{\mu}_{\text{CF}} = \mu_{\text{CF}} - \Sigma_{\text{CF}}^0$, thus leading to

$$\Gamma_{\text{SC}}(\mathbf{P},\Omega) \approx \frac{2\pi\varepsilon_0}{m_r} \frac{1}{i\Omega - \frac{P^2}{2M} + \tilde{\mu}_{\text{CF}}} \,. \tag{23}$$

$\Gamma_{\text{SC}}(\mathbf{P},\Omega)$ then appears to be proportional to a bare-like fermionic Green's function,

$$\tilde{G}_{\text{CF}}^0(\mathbf{P},\Omega) \equiv \frac{1}{i\Omega - P^2/2M + \tilde{\mu}_{\text{CF}}} \,, \tag{24}$$

of a particle of mass $M$ with bare dispersion $P^2/2M - \mu_{\text{CF}}$ renormalized by a mean-field shift $\Sigma_{\text{CF}}^0$ due to interaction with the atomic medium, the constant of proportionality $2\pi\varepsilon_0/m_r$ in Eq. (20) originating from the composite nature of the molecules.

Atomic fermions can also correlate with bosons belonging to the condensate during a pair-breaking event (see Fig. 1(b)); this process is taken into account in the many-body $T$-matrix in the condensed phase $T(\mathbf{P},\Omega)$ as a self-energy insertion in the related Dyson's (or Bethe-Salpeter) equation (8) which in the strong coupling limit acquires the form

$$T_{\text{SC}}^{-1}(\mathbf{P},\Omega) = \Gamma_{\text{SC}}^{-1}(\mathbf{P},\Omega) - n_0 G_F^0(\mathbf{P},\Omega) \tag{25}$$

$$= \frac{m_r}{2\pi\varepsilon_0} \left( i\Omega - \frac{P^2}{2M} + \tilde{\mu}_{\text{CF}} - \frac{\frac{2\pi\varepsilon_0 n_0}{m_r}}{i\Omega - \frac{P^2}{2m_F} + \mu_F} \right) \tag{26}$$

$$\equiv \frac{m_r}{2\pi\varepsilon_0} G_{\text{CF}}^{-1}(\mathbf{P},\Omega). \tag{27}$$

Here, $G_{\text{CF}}(\mathbf{P},\Omega)$, as defined by Eq. (27), can be interpreted as a dressed composite fermion propagator, resulting from the hybridization between an undressed molecular bound state, described by $\tilde{G}_{\text{CF}}^0(\mathbf{P},\Omega)$, and an unpaired state made of a fermionic atom and a boson belonging to the condensate.

Specifically, the quantity

$$\Delta_0^2 \equiv 2\pi\varepsilon_0 n_0 / m_r \tag{28}$$

naturally arises as an energy scale associated with the exchange of bosons with the condensate and reveals its meaning through an analysis of the pole structure of $T_{\text{SC}}(\mathbf{P},\Omega)$, by which one finds the dispersion relations

$$E_{\mathbf{P}}^{\pm} = \frac{\tilde{\xi}_{\mathbf{P}}^{\text{CF}} + \xi_{\mathbf{P}}^F \pm \sqrt{(\tilde{\xi}_{\mathbf{P}}^{\text{CF}} - \xi_{\mathbf{P}}^F)^2 + 4\Delta_0^2}}{2} \,. \tag{29}$$

They denote the hybridization of a molecule with renormalized dispersion $\tilde{\xi}_{\mathbf{P}}^{\text{CF}} = P^2/2M - \tilde{\mu}_{\text{CF}}$ with an unpaired atom with dispersion $\xi_{\mathbf{P}}^{\text{F}} = P^2/2m_{\text{F}} - \mu_{\text{F}}$ and a boson in the condensate with vanishing kinetic energy. This kind of hybridization was first found in [65] within a two-channel model.

One sees from Eq. (29) that $\Delta_0$ controls the amount of hybridization between $\tilde{\xi}_{\mathbf{P}}^{\text{CF}}$ and $\xi_{\mathbf{P}}^{\text{F}}$ and that, for small $\Delta_0$, the hybridized dispersions $E_{\mathbf{P}}^{\pm}$ tend either to $\tilde{\xi}_{\mathbf{P}}^{\text{CF}}$ or $\xi_{\mathbf{P}}^{\text{F}}$ depending on the sign of $\tilde{\xi}_{\mathbf{P}}^{\text{CF}} - \xi_{\mathbf{P}}^{\text{F}}$, as it will be detailed in section 3.4. We note in this respect that the scale $\Delta_0$ vanishes when the product $\varepsilon_0 n_0 \to 0$. If, like in 3D, $n_0$ would vanish identically above a certain coupling strength for all $x \leq 1$, then one would obtain $\Delta_0 = 0$ in the strong-coupling limit for all concentrations considered in the present work. However, as already mentioned, $n_0$ vanishes only exponentially with the coupling strength in our approach, and the presence of the exponentially large factor $\varepsilon_0$ in the definition of $\Delta_0$ can compensate for the vanishing of $n_0$. We will see indeed below that, in the strong-coupling limit, $\Delta_0 \to 0$ only when $x \to 0$ or $x \to 1$, with $\Delta_0$ tending to a finite (non-vanishing) value for all intermediate concentrations.

The composite-fermion propagator $G_{\text{CF}}(\mathbf{P}, \Omega)$ acquires a simple form when expressed in terms of the quasi-particle energies $E_{\mathbf{P}}^{\pm}$ and corresponding weights $u_{\mathbf{P}}^2$ and $v_{\mathbf{P}}^2$,

$$G_{\text{CF}}(\mathbf{P}, \Omega) = \frac{u_{\mathbf{P}}^2}{i\Omega - E_{\mathbf{P}}^+} + \frac{v_{\mathbf{P}}^2}{i\Omega - E_{\mathbf{P}}^-}, \tag{30}$$

with

$$u_{\mathbf{P}}^2 = \frac{1}{2}\left(1 + \frac{\tilde{\xi}_{\mathbf{P}}^{\text{CF}} - \xi_{\mathbf{P}}^{\text{F}}}{\sqrt{(\tilde{\xi}_{\mathbf{P}}^{\text{CF}} - \xi_{\mathbf{P}}^{\text{F}})^2 + 4\Delta_0^2}}\right), \qquad \text{and} \qquad v_{\mathbf{P}}^2 = 1 - u_{\mathbf{P}}^2. \tag{31}$$

The number of composite fermions $n_{\text{CF}}$ is swiftly obtained for given coupling and population imbalance as

$$n_{\text{CF}} = \int \frac{d\mathbf{P}}{(2\pi)^2} \int \frac{d\Omega}{2\pi} G_{\text{CF}}(\mathbf{P}, \Omega) e^{i\Omega 0^+} \tag{32}$$

$$= \int \frac{d\mathbf{P}}{(2\pi)^2} \left[u_{\mathbf{P}}^2 \Theta(-E_{\mathbf{P}}^+) + v_{\mathbf{P}}^2 \Theta(-E_{\mathbf{P}}^-)\right], \tag{33}$$

where the integrand in Eq. (33) can be interpreted as the composite-fermion momentum distribution function $n_{\text{CF}}(\mathbf{P})$. We further note that, in practice, only the term containing $v_{\mathbf{P}}^2$ contributes in Eq. (33) since $E_{\mathbf{P}}^+$ turns out to be always positive for the concentrations and couplings considered in our work. One thus has

$$n_{\text{CF}}(\mathbf{P}) \equiv \left[u_{\mathbf{P}}^2 \Theta(-E_{\mathbf{P}}^+) + v_{\mathbf{P}}^2 \Theta(-E_{\mathbf{P}}^-)\right] = v_{\mathbf{P}}^2 \Theta(-E_{\mathbf{P}}^-). \tag{34}$$

We finally note that the integration over $\mathbf{P}$ in Eq. (33) can be performed in a closed form (see Eq. (D.1) of appendix D).

In conclusion, we have found that molecular BF states are: i) renormalized by the presence of the remaining unpaired fermions via the mean-field self-energy $\Sigma_{\text{CF}}^0$, and ii) hybridized with atomic states owing to pair-breaking fluctuations that foster a non-vanishing condensate. For small bosonic concentrations, the latter feature can also be viewed as feedback of polaronic correlations on the many-body $T$-matrix $T(\mathbf{P}, \Omega)$.

## 3.2 Fermionic self-energy and momentum distribution

We now pass to examine the fermionic self-energy Eq. (12) in the strong-coupling limit, that is, with $T(\mathbf{P}, \Omega)$ replaced by $T_{\text{SC}}(\mathbf{P}, \Omega)$.

The frequency integration in Eq. (12) is computed through a contour integration on the left-hand side of the complex plane, due to the presence of the convergence factor $e^{i\omega 0^+}$. The pole originating from $G_{\mathrm{B}}^0(\mathbf{P}-\mathbf{k}, \Omega-\omega)$ is located on the right-hand side of the complex plane because $\mu_{\mathrm{B}} < 0$, and does not contribute to the integral. The contributions from the poles of $T_{\mathrm{SC}}$ yield instead

$$\Sigma_{\mathrm{F}}(\mathbf{k},\omega) = \frac{\Delta_0^2}{i\omega - \tilde{\xi}_{\mathbf{k}}^{\mathrm{CF}}} + \left(\frac{2\pi\varepsilon_0}{m_r}\right)\int\frac{d\mathbf{P}}{(2\pi)^2}\left[\frac{u_{\mathbf{P}}^2\Theta(-E_{\mathbf{P}}^+)}{i\omega + \xi_{\mathbf{P}-\mathbf{k}}^{\mathrm{B}} - E_{\mathbf{P}}^+} + \frac{v_{\mathbf{P}}^2\Theta(-E_{\mathbf{P}}^-)}{i\omega + \xi_{\mathbf{P}-\mathbf{k}}^{\mathrm{B}} - E_{\mathbf{P}}^-}\right]. \tag{35}$$

In the strong-coupling limit and for $x \leq 1$, the boson chemical potential $\mu_{\mathrm{B}} \simeq -\varepsilon_0$ is the dominant energy scale. One is then allowed to take $\mathbf{P} = 0$ in the denominators of the integrand of Eq. (35) and neglect $E_{\mathbf{P}=0}^{\pm}$, which is of order $E_{\mathrm{F}}$, with respect to $\mu_{\mathrm{B}}$ therein, thus obtaining

$$\Sigma_{\mathrm{F}}(\mathbf{k},\omega) = \frac{\Delta_0^2}{i\omega - \tilde{\xi}_{\mathbf{k}}^{\mathrm{CF}}} + \frac{\frac{2\pi\varepsilon_0}{m_r}n_{\mathrm{CF}}}{i\omega + \xi_{\mathbf{k}}^{\mathrm{B}}}, \tag{36}$$

where $n_{\mathrm{CF}}$ is given by Eq. (34).

The large energy scale $\varepsilon_0$ appearing in the second term on the r.h.s of Eq. (36) is compensated by $\mu_{\mathrm{B}}$ in the denominator, thus making the two terms in Eq. (36) of the same order. By conveniently introducing the energy scale

$$\Delta_{\mathrm{CF}}^2 = \frac{2\pi\varepsilon_0 n_{\mathrm{CF}}}{m_r}, \tag{37}$$

the fermionic Green's function reads

$$G_{\mathrm{F}}^{-1}(\mathbf{k},\omega) = i\omega - \xi_{\mathbf{k}}^{\mathrm{F}} - \frac{\Delta_0^2}{i\omega - \tilde{\xi}_{\mathbf{k}}^{\mathrm{CF}}} - \frac{\Delta_{\mathrm{CF}}^2}{i\omega + \xi_{\mathbf{k}}^{\mathrm{B}}}. \tag{38}$$

An analogous form was obtained in [111,112], albeit in the normal phase and in 3D.

The Green's function (38), when integrated over the frequency as in Eq. (15), determines the fermion momentum distribution $n_{\mathrm{F}}(\mathbf{k})$. A simple contour integration then yields

$$n_{\mathrm{F}}(\mathbf{k}) = \sum_{i=1}^{3}\lim_{z\to z_i}\frac{z - z_i}{z - \xi_{\mathbf{k}}^{\mathrm{F}} - \frac{\Delta_0^2}{z-\tilde{\xi}_{\mathbf{k}}^{\mathrm{CF}}} - \frac{\Delta_{\mathrm{CF}}^2}{z+\xi_{\mathbf{k}}^{\mathrm{B}}}}\Theta(-z_i), \tag{39}$$

where $z_i$, $i = 1, 2, 3$ are the three different (real) roots of the cubic equation

$$z - \xi_{\mathbf{k}}^{\mathrm{F}} - \frac{\Delta_0^2}{z - \tilde{\xi}_{\mathbf{k}}^{\mathrm{CF}}} - \frac{\Delta_{\mathrm{CF}}^2}{z + \xi_{\mathbf{k}}^{\mathrm{B}}} = 0. \tag{40}$$

One root is essentially given by $z = -\xi_{\mathbf{k}}^{\mathrm{B}}$, while the other two are roughly given by $z = \tilde{\xi}_{\mathbf{k}}^{\mathrm{CF}}$ and $z = \xi_{\mathbf{k}}^{\mathrm{F}}$.

A more explicit expression for $n_{\mathrm{F}}(\mathbf{k})$ can be obtained when the hybridization energy $\Delta_0 \ll E_{\mathrm{F}}$. Within our theoretical approach, this condition occurs either when $x \to 1$ or when $x \to 0$. One obtains (details are provided in Appendix D.2)

$$n_{\mathrm{F}}(\mathbf{k}) = n_{\mathrm{CF}}|\phi(\mathbf{k})|^2 + n_{\mathrm{UF}}(\mathbf{k}), \tag{41}$$

where

$$\phi(\mathbf{k}) = \sqrt{\frac{2\pi\varepsilon_0}{m_r}}\frac{1}{\frac{\mathbf{k}^2}{2m_r} + \varepsilon_0}, \tag{42}$$

is the bound-state wave-function for the two-body problem in vacuum, while

$$n_{\text{UF}}(\mathbf{k}) = \left[ 1 - n_{\text{CF}} |\phi(\mathbf{k})|^2 - \frac{\Delta_0^2 \Theta(\tilde{\xi}_{\mathbf{k}}^{\text{CF}})}{(\tilde{\xi}_{\mathbf{k}}^{\text{CF}} - \tilde{\xi}_{\mathbf{k}}^{\text{F}})^2} \right] \Theta(-\tilde{\xi}_{\mathbf{k}}^{\text{F}}) + \frac{\Delta_0^2 \Theta(-\tilde{\xi}_{\mathbf{k}}^{\text{CF}})}{(\tilde{\xi}_{\mathbf{k}}^{\text{CF}} - \tilde{\xi}_{\mathbf{k}}^{\text{F}})^2} \Theta(\tilde{\xi}_{\mathbf{k}}^{\text{F}}),$$ (43)

with $\tilde{\xi}_{\mathbf{k}}^{\text{F}} \equiv \xi_{\mathbf{k}}^{\text{F}} + 2\pi n_{\text{CF}}/m_r$. The first term in Eq. (41) corresponds to fermions bound in molecules, while the second one corresponds to unpaired fermions, as discussed in Appendix D.2.

### 3.3 Hugenholtz-Pines condition for condensed bosons

In the absence of boson-boson repulsion ($\Sigma_{11} = \Sigma_{12} = 0$) and in the strong-coupling limit of interest here, the bosonic self-energy (11) acquires the form

$$\Sigma_{\text{B}}(\mathbf{k}, \omega) = \left( \frac{2\pi\varepsilon_0}{m_r} \right) \int \frac{d\mathbf{P}}{(2\pi)^2} \left[ u_{\mathbf{P}}^2 \frac{\Theta(-E_{\mathbf{P}}^+) - \Theta(-\xi_{\mathbf{P}-\mathbf{k}}^{\text{F}})}{E_{\mathbf{P}}^+ - \xi_{\mathbf{P}-\mathbf{k}}^{\text{F}} - i\omega} + v_{\mathbf{P}}^2 \frac{\Theta(-E_{\mathbf{P}}^-) - \Theta(-\xi_{\mathbf{P}-\mathbf{k}}^{\text{F}})}{E_{\mathbf{P}}^- - \xi_{\mathbf{P}-\mathbf{k}}^{\text{F}} - i\omega} \right],$$ (44)

when $T(\mathbf{P}, \Omega)$ is replaced by $T_{\text{SC}}(\mathbf{P}, \Omega)$ in Eq. (9) and the frequency integral is performed.

The above integral can be solved analytically when, as required in the Hugenholtz-Pines condition (19), it is evaluated at zero momentum and frequency, as shown in Appendix D.

### 3.4 Complete system of semi-analytic equations in closed form and their solutions

Collecting the analytic results obtained so far, it turns out that the system of Eqs. (17)-(19) becomes analytically tractable in the strong-coupling limit. Equations (17) and (19) are given in closed form in Eq. (39) and in Eq. (D.4) of Appendix D, respectively. Equation (18) can be dealt with by observing that for large $g$ all non-condensed bosons are expected to bind with fermions into molecules (we recall that $n_{\text{B}} \leq n_{\text{F}}$). One can then substitute Eq. (18) for bosons with Eq. (33) for composite fermions since $n_{\text{B}} - n_0 \simeq n_{\text{CF}}$. As a consequence, one obtains the following set of equations for the unknowns $\mu_{\text{F}}, \mu_{\text{B}}$ and $n_0$ (and implicitly $\mu_{\text{CF}} \equiv \mu_{\text{F}} + \mu_{\text{B}} + \varepsilon_0$):

- Fermion number equation

$$n_{\text{F}} = \int \frac{d\mathbf{k}}{(2\pi)^2} n_{\text{F}}(\mathbf{k}),$$ (45)

with $n_{\text{F}}(\mathbf{k})$ given by Eq. (39);

- Composite-fermion number Eq. (33)

$$n_{\text{B}} - n_0 = \int \frac{d\mathbf{P}}{(2\pi)^2} \left[ u_{\mathbf{P}}^2 \Theta(-E_+) + v_{\mathbf{P}}^2 \Theta(-E_-) \right],$$ (46)

whose analytic form is given in Eq. (D.1);

- Hugenholtz-Pines condition (19)

$$\mu_{\text{B}} = \Sigma_{\text{B}}(\mathbf{0}, 0) = \left( \frac{2\pi\varepsilon_0}{m_r} \right) \int \frac{d\mathbf{P}}{(2\pi)^2} \left[ \frac{u_{\mathbf{P}}^2}{E_{\mathbf{P}}^+ - \xi_{\mathbf{P}}^{\text{F}}} \Theta(-E_{\mathbf{P}}^+) + \frac{v_{\mathbf{P}}^2}{E_{\mathbf{P}}^- - \xi_{\mathbf{P}}^{\text{F}}} \Theta(-E_{\mathbf{P}}^-) \right],$$ (47)

whose analytic expression is given in Eq. (D.4).

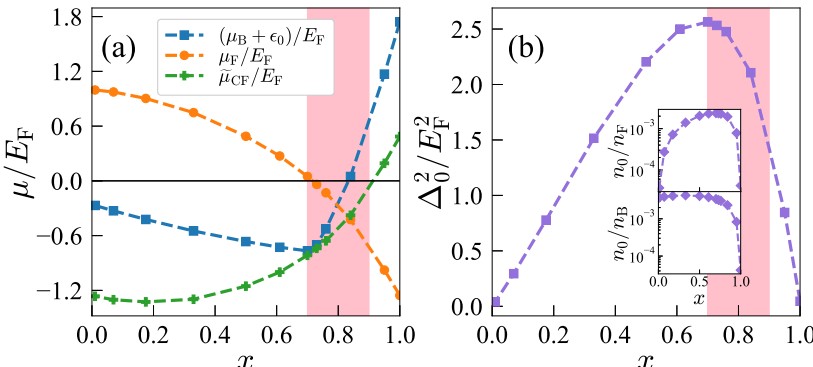

Figure 3: (a): Chemical potentials $\mu_F$, $\mu_B + \varepsilon_0$ and $\tilde{\mu}_{CF} = \mu_{CF} - \Sigma_{CF}^0$ (in units of $E_F$) as functions of concentration $x$, for strong BF attraction $g = 2.8$ and BB repulsion $\eta_B = 0$. (b): Square of the hybridization energy $\Delta_0^2 = 2\pi\varepsilon_0 n_0/m_r$ (in units of $E_F^2$) as a function of the boson concentration $x$, for strong BF attraction $g = 2.8$ and BB repulsion $\eta_B = 0$. Insets: Condensate density $n_0$ in units of $n_F$ (top) and condensate fraction $x$ (bottom) for the same set of parameters.

This simple system of equations can be solved for $\mu_F, \mu_B$, and $n_0$ with a standard root finder, and no particular care has to be taken when performing the related momentum integrals. Figures 3–6 present the results for different physical quantities based on the solutions of Eqs. (45)-(47). In all calculations, we will consider equal boson and fermion masses: $m_B = m_F$.

In Fig. 3(a) the fermion chemical potential $\mu_F$ and the boson chemical potential $\mu_B$ (once subtracted the leading term $-\varepsilon_0$) are shown along with the shifted composite fermion chemical potential $\tilde{\mu}_{CF} = \mu_{CF} - \Sigma_{CF}^0$. Trivially, the bosonic chemical potential $\mu_B$ follows the dominant energy scale of the binding energy $-\varepsilon_0$, implying that as soon as a boson is added to the system it binds with a fermion (recalling that $n_B \leq n_F$).

Two clearly different regimes can be distinguished in the opposite limits $x \to 0$ and $x \to 1$. For $x \simeq 0$, the system is made of weakly interacting atomic fermions and composite fermions (molecules) which are strongly affected by interaction. One sees indeed that $\mu_F$ approaches the non-interacting value $E_F$. At the same time, $\tilde{\mu}_{CF}$ is negative, implying that $\Theta(-\tilde{\xi}_P^{CF}) = 0$ and a non-vanishing composite fermion density is obtained only because the composite fermion momentum distribution $n_{CF}(\mathbf{P}) = v_{\mathbf{P}}^2 \Theta(-E_{\mathbf{P}}^-)$ strongly differs from the ideal Fermi distribution $\Theta(-\tilde{\xi}_{\mathbf{P}}^{CF})$ (see also Fig. 5 below), indicating strong interaction effects on the molecules. The reverse situation occurs for $x \to 1$. In this case $\mu_F < 0$, implying that $\Theta(-\xi_{\mathbf{k}}^F) = 0$, while $\tilde{\mu}_{CF}$ approaches $E_F/2$, the expected value for a filled Fermi sphere of non-interacting molecules of mass $M = 2m_F$ (note also that $\Sigma_{CF}^0 = 0$ in this case, implying $\tilde{\mu}_{CF} = \mu_{CF}$).

We therefore identify a crossover region $0.7 \lesssim x \lesssim 0.9$ (shaded area in Fig. 3) in which the two chemical potentials $\mu_F$ and $\tilde{\mu}_{CF}$ cross and change their sign (somewhat analogously with the use of the chemical potential $\mu_F$ as a proxy for the crossover region when studying the BCS-BEC crossover [113, 114]). In this region, atoms and molecules are in a state of maximal hybridization as reflected in Fig. 3(b) reporting the square of the hybridization energy $\Delta_0^2/E_F^2 = 2\pi\varepsilon_0 n_0/m_r$ introduced in Sec. 3.1.

In the inset of Fig. 3(b) (upper panel) the condensate density $n_0$ is also displayed, to show that its value is exponentially suppressed for strong coupling. This exponentially small value is however compensated by the exponentially large binding energy in the expression for $\Delta_0$, yielding the sizable values of $\Delta_0$ shown in the main panel of Fig. 3(b). For completeness, the inset of Fig. 3(b) (lower panel) also reports the condensate fraction $x$ as a function of concentration for the same strong BF attraction.

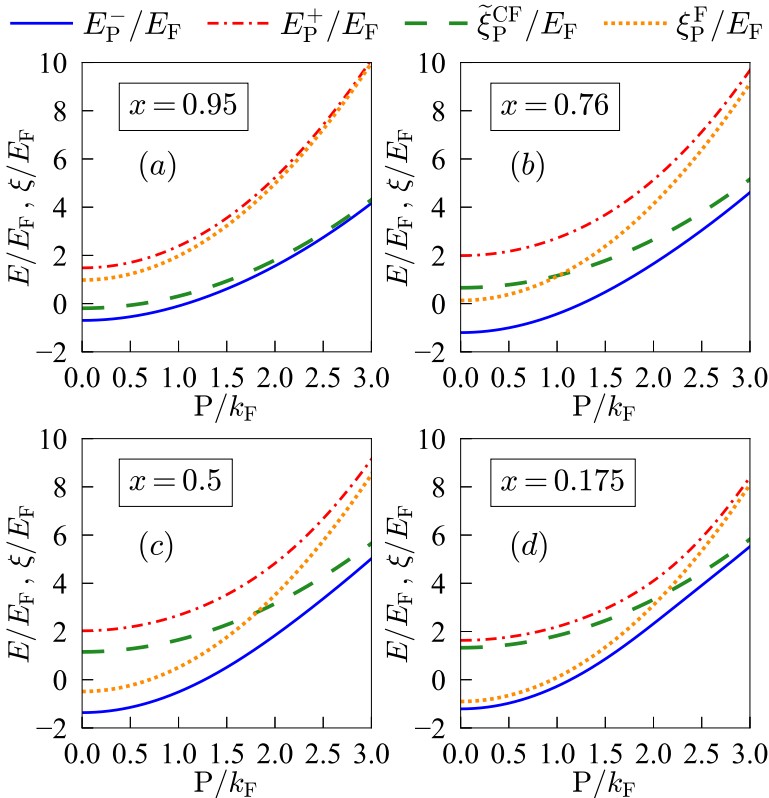

Figure 4: Hybridized dispersions $E_{\mathbf{P}}^{\pm}$ of the poles of the strong-coupling limit $T$-matrix $T_{\mathrm{SC}}(\mathbf{P},\Omega)$ and unhybridized molecular and atomic dispersions $\tilde{\xi}_{\mathbf{P}}^{\mathrm{CF}}$ and $\xi_{\mathbf{P}}^{\mathrm{F}}$ (in units of $E_{\mathrm{F}}$) as function of $P/k_{\mathrm{F}}$. All data are obtained for strong BF attraction $g = 2.8$ and BB repulsion $\eta_{\mathrm{B}} = 0$. Different bosonic concentrations are considered: (a) $x = 0.95$, (b) $x = 0.76$, (c) $x = 0.5$, (d) $x = 0.175$.

Let us now analyze in detail the effect of the hybridization scale $\Delta_0$ through the energy spectrum $E_{\mathbf{P}}^{\pm}$ of the molecular quasi-particles defined by Eq. (29) of Sec. 3.1. In Fig. 4 the hybridized dispersions $E_{\mathbf{P}}^{\pm}$ are compared with those of the unhybridized molecular and atomic fermions for coupling $g = 2.8$ and different values of the boson concentration $x$. At exactly matched densities ($x = 1$) all fermions are essentially paired with all bosons ($n_0 \approx 0$) and the system is effectively made of a gas of non-interacting molecules, with the unhybridized dispersions coinciding with the hybridized ones since $\Delta_0 \simeq 0$.

As soon as the concentration decreases (see the case $x = 0.95$ in Fig. 4(a)), the hybridization energy scale $\Delta_0$ sharply rises and hybrid quasi-particles form out of the molecular and unpaired Fermi states. As the concentration further decreases, even though $n_0$ remains exponentially suppressed, the presence of a non-zero $\Delta_0 \propto \sqrt{n_0 \varepsilon_0}$ shifts down the energy dispersion $E_{\mathbf{P}}^{-}$ with respect to the unhybridized dispersion $\tilde{\xi}_{\mathbf{P}}^{\mathrm{CF}}$. This effect is rooted in the kinetic energy cost when filling up the paired and unpaired Fermi spheres: in order to reduce it, the molecular and atomic wave-functions hybridize quantum-mechanically by overlapping with the condensate wave-function (thus gaining in delocalization energy). The bottom panels of Fig. 4 display the typical level crossing (dotted and dashed lines) and avoided crossing (solid and dot-dashed lines) of a non-hybridized and hybridized two-level system, respectively. At the crossover concentration $x \simeq 0.7$, the hybridization is about maximal and the character of the occupied states (bottom pole dispersion) switches from molecular to mostly atomic as $x$ is reduced (with an effective mass approaching $m_{\mathrm{F}}$). At small concentration ($x = 0.175$),

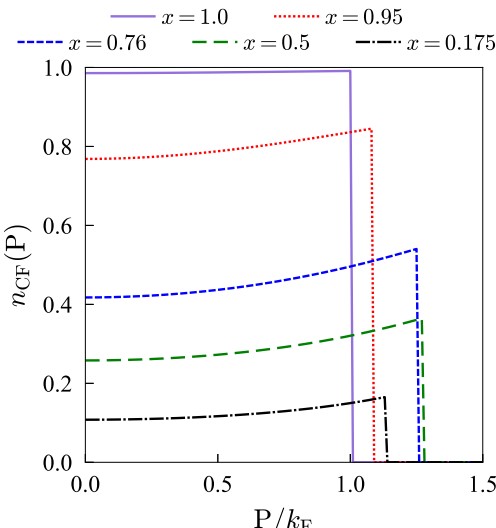

Figure 5: Composite-fermion momentum distribution $n_{\mathrm{CF}}(P) = v_{\mathbf{P}}^2 \Theta(-E_{\mathbf{P}}^-)$ as a function of $P/k_{\mathrm{F}}$, for BF coupling $g = 2.8$, BB repulsion $\eta_{\mathrm{B}} = 0$, and concentrations matching those of Fig. 4, along with the case $x = 1$.

the dispersion of occupied composite fermion states $E_{\mathbf{P}}^-$ approaches the non-interacting atomic dispersion $\xi_{\mathbf{P}}^{\mathrm{F}}$, albeit with a small quasi-particle weight $v_{\mathbf{P}}^2 \simeq 0$ (see also Fig. 5). So, in this regime, the occupied composite fermion states are determined by the small participation of molecules to the hybridized dispersion $E_{\mathbf{P}}^-$ with a predominant atomic fermion character.

The momentum distribution of the composite fermions $n_{\mathrm{CF}}(P) = v_{\mathbf{P}}^2 \Theta(-E_{\mathbf{P}}^-)$ is shown in Fig. 5 for a number of concentrations corresponding to those of Fig. 4 along with the case $x = 1$. In the latter case, the mixture is essentially made of a single-component gas of non-interacting point-like fermionic molecules, as illustrated by the Fermi step behavior with occupation number approaching unity. As $x$ decreases, interaction effects, originating from the hybridization of the molecular and atomic states, become significant and generate a rather peculiar behavior of the momentum distribution function, as evidenced by its upward bending with increasing momentum. (A similar behavior was also found in [71] for a BF mixture in a 1D lattice.) Moreover, we notice that the Fermi step, in particular for small concentrations, is located well above the expected position if one were to assume the validity of the Luttinger theorem for the composite fermions.

The Luttinger theorem states that the volume of the Fermi sphere of a Fermi liquid does not depend on interaction, and thus coincides with the volume of the Fermi sphere of the corresponding non-interacting Fermi gas [115]. Its validity has been extended to density-imbalanced two-component Fermi gases [116] and, within a two-channel model, to BF mixtures [65]. In both cases, it was shown that, provided the system is in the normal phase, the volumes of the Fermi spheres are separately conserved for the two Fermi components (in the BF mixture case, the two components correspond to renormalized fermionic atoms and molecules). This is indeed what is found in the strong-coupling limit of a 3D BF mixture in [84], since in this limit the condensate vanishes identically. In contrast, in the present 2D case the condensate density never identically vanishes, and in particular the hybridization energy $\Delta_0$ remains finite even for $g \to \infty$ (except in the trivial case $x = 0$ and in the more interesting case $x = 1$).

In a BF mixture with a condensate, it was shown in [65] that it is the sum of the volumes of the Fermi spheres of dressed atomic fermions and molecules that should be conserved. Fig. 5 shows a clear violation of the latter version of the Luttinger theorem since the molecular Fermi

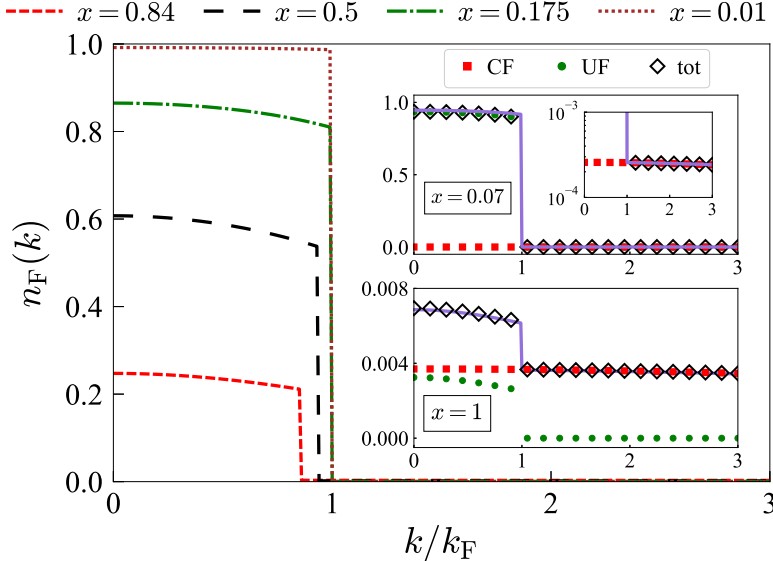

Figure 6: Fermionic momentum distribution $n_F(\mathbf{k})$ as a function of $k/k_F$, as obtained from the strong-coupling asymptotic expression (39), for various bosonic concentrations $x$, at BF coupling $g = 2.8$ and BB repulsion $\eta_B = 0$. In the insets, the same expression (full line) is compared with the analytical contributions obtained in the limit of small hybridization: $n_{CF}|\phi(\mathbf{k})|^2$ (CF) and $n_{UF}(\mathbf{k})$ (UF) as given by Eq. (43), and their sum (tot).

momentum is always larger than $k_F$, which is the Fermi momentum that should determine the *total* volume (i.e the sum of the volumes of the Fermi spheres of dressed atomic fermions and molecules). We attribute this shortcoming to two factors. First, the large values of the hybridization energy-scale $\Delta_0$ obtained in our theory, which are in turn caused by an overestimation of the atom-molecule repulsion (see Sec. 3.1), eventually lead to an overpopulation of the condensate. So, the simple strong-coupling picture of a Fermi-Fermi mixture of unpaired fermions and molecules obeying the Luttinger theorem is never reached in 2D. Second, as stressed in [116] and explicitly shown in [117], when dealing with approximate theories the validity of the Luttinger theorem is not guaranteed for schemes, like the present one, that do not use fully self-consistent fermionic Green's functions in their formulation.

Nevertheless, the breakdown of the Luttinger theorem in 2D for the hybridized composite fermions is compensated, as far as the integrated quantity $n_{CF} = \int \frac{d\mathbf{P}}{(2\pi)^2} n_{CF}(\mathbf{P}) = \int \frac{d\mathbf{P}}{(2\pi)^2} v_{\mathbf{P}}^2 \Theta(-E_{\mathbf{P}}^-)$ is concerned, by the strong suppression of the weight $v_{\mathbf{P}}^2$ when the concentration decreases, so that the identification of $n_{CF}$ with $n_B - n_0 \simeq n_B$ is always valid in the strong-coupling limit.

Finally, we discuss the profiles of the Fermi momentum distribution (39) for fixed coupling strength and different concentrations. In Fig. 6, $n_F(\mathbf{k})$ is displayed for $g = 2.8$ and different values of $x$. For small $x$, most fermions remain unpaired and are described by a weakly renormalized Fermi step, while a small fraction of fermions bound in molecules are described by the small contribution $n_{CF}|\phi(\mathbf{k})|^2$, with $n_{CF} \ll n_F$. As $x$ increases, the height of the Fermi step progressively reduces, indicating the increasing importance of interaction effects. The position of the Fermi step, on the other hand, decreases only slightly from the non-interacting value $k_F$. We notice that this behavior, together with the one just discussed for $n_{CF}(\mathbf{P})$ reflects again the breakdown of the Luttinger theorem that, in its modified version in the presence of a condensate [65], would require $k_F^2 = k_{F,UF}^2 + P_{F,CF}^2$ (where $k_{F,UF}$ and $P_{F,CF}$ indicate the Fermi momenta for the unpaired and composite fermions, respectively).

An important exception is the case $x = 1$. In this case, the height of the Fermi step for $n_F(\mathbf{k})$ is given by the quasi-particle weight $\Delta_0^2/(\tilde{\xi}_{\mathbf{k}}^{\mathrm{CF}} - \tilde{\xi}_{\mathbf{k}})^2$ (cf. Eq. (D.22b) of Appendix D.2), with $\Delta_0$ vanishing in the limit $g \to \infty$ (see also Fig. 15 (b) below). The Fermi step thus vanishes, effectively corresponding to $k_{F,UF} = 0$, and $n_F(\mathbf{k})$ is entirely described by the contribution $n_{CF}|\phi(\mathbf{k})|^2$ from fermions bound into molecules. At the same time, $P_{F,CF} = k_F$, as one can see in Fig. 5, consistently with the Luttinger theorem for a system made only by molecules.

The insets of Fig. 6 finally compare the full strong-coupling expression (39) for $n_F(\mathbf{k})$ with its small $\Delta_0/E_F$ expansion, Eqs.(41-43), which holds for $x \to 0$ and $x \to 1$, as mentioned above. The distribution function (41) resulting from this expansion (tot) is broken down into paired (CF) and unpaired (UF) contributions, described by $n_{CF}|\phi(\mathbf{k})|^2$ and Eq. (43) for $n_{UF}(\mathbf{k})$, respectively. From this comparison, one sees that the small $\Delta_0/E_F$ expansion fully agrees with the full strong-coupling expression (39). One also sees how, for $x = 1$, the unpaired contribution is already strongly suppressed for the value $g = 2.8$ considered in Fig. 6, consistently with its vanishing contribution for $g \to \infty$ discussed above.

# 4 Numerical results

In this section, we present the fully numerical results for the boson and fermion momentum distributions and for key thermodynamic quantities like the chemical potentials and condensate fraction with varying coupling strength. We also discuss the results for Tan's contact parameter [118–120]. In all calculations, we consider only the case of equal boson and fermion masses: $m_B = m_F$.

## 4.1 Methods

We recall the system of equations constituting the bulk of our numerical calculations

$$\mu_B = \Sigma_B(\mathbf{k} = \mathbf{0}, \omega = 0) - \Sigma_{12}, \tag{48a}$$

$$n_F = \int \frac{d\mathbf{k}}{(2\pi)^2} n_F(\mathbf{k}), \tag{48b}$$

$$n_B = n_0 + n_B' = n_0 + \int \frac{d\mathbf{k}}{(2\pi)^2} n_B(\mathbf{k}). \tag{48c}$$

For given values of $n_F, n_B, a_{BF}$ and $a_{BB}$ the above system is solved for the unknowns $\mu_F, \mu_B$ and $n_0$, as follows. First, a solution is found for $\mu_B$ from Eq. (48a) with a standard bisection method assuming a suitable ansatz for $\mu_F$ and $n_0$. Then we consider the 2×2 system of the remaining Eqs. (48b) and (48c) as functions of $\mu_F$ and $n_0$ and apply a two-dimensional secant (quasi-Newton) method whereby the approximate Jacobian matrix (which corresponds to the approximate Hessian of the total energy) is updated according to a symmetric rank 1 algorithm [121], which is a generalization of the secant method to multidimensional problems.

An intermediate step of the above procedure is the evaluation of the bosonic and fermionic self-energies, which requires, owing to the low dimensionality of the problem at hand, special care when dealing with the slow convergence of the frequency convolutions appearing in Eqs. (11)-(12). In order to speed up the convergence, the integrand of the convolutions is added and subtracted by an auxiliary function with the same asymptotic behavior yet analytically integrable. As a result, the numerical integration is truncated by applying a large frequency cutoff, which we fix at $\Omega_c = \pm 50000 E_F$ after an accurate analysis of the asymptotic behaviors. For $|\Omega| \geq \Omega_c$ integration is done by making use of asymptotic expressions whose details are provided in Appendix C. In addition, integration in momentum space is affected by

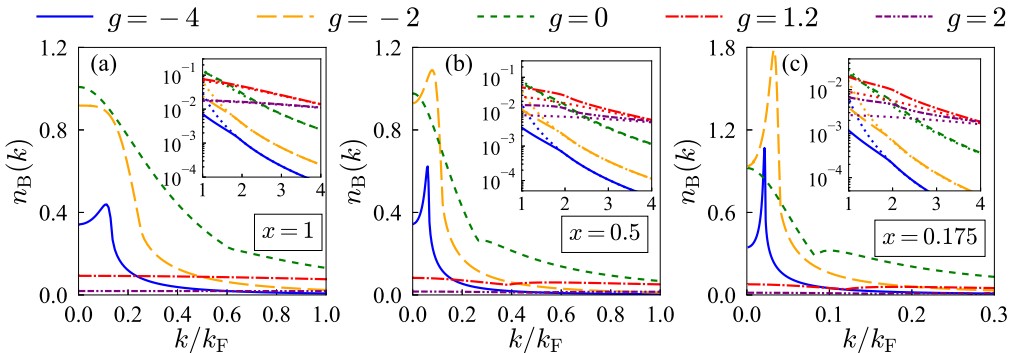

Figure 7: Bosonic momentum distribution $n_B(k)$ as a function of $k/k_F$, for different values of BF coupling $g$ and BB repulsion $\eta_B = 0$. Different cases of bosonic concentration $x$ are reported: (a) $x = 1$, (b) $x = 0.5$, (c) $x = 0.175$. Insets: comparison between numerical results and large momentum behavior described by Eq. (49) (dotted lines).

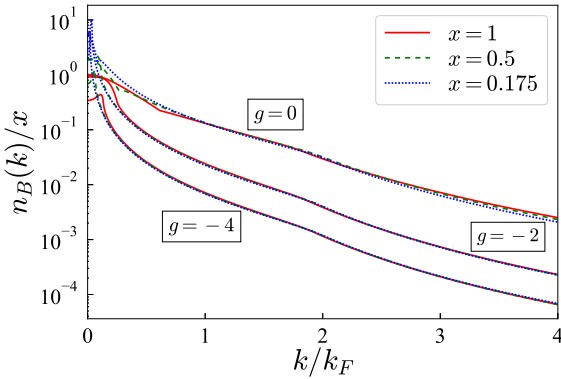

Figure 8: Bosonic momentum distribution $n_B(k)$ divided by the boson concentration $x$ as a function of $k/k_F$, for several values of $x$ and BF coupling $g$.

the presence of discontinuities in the integrand due to Fermi steps. Therefore, a careful analysis of the location of these discontinuities is performed to identify the appropriate integration intervals, for which a standard Gauss-Legendre quadrature method is adopted [122]. The largest Fermi momentum provides a natural cutoff for the momentum convolutions defining both bosonic and fermionic self-energies.

Concerning the frequency integral yielding the momentum distributions (15) and (16) we take advantage of the large frequency behavior of the self-energies (C.5-C.6) valid in the range $|\omega| \geq 100 E_F$, whereas for $|\omega| < 100 E_F$ we integrate the full numerical expression (see Appendix C.2). Momentum integration in expressions (17) and (18) is carried out numerically up to a cutoff of $k_c = 4k_F$, after having subtracted and added a non-interacting or a Bogoliubov Green's function respectively (see Appendix C.2), and beyond $k_c$ by making use of the asymptotic expressions (60-61).

## 4.2 Boson and fermion momentum distributions

We now discuss the numerical results for the fermionic and bosonic momentum distributions, Eqs. (15)-(16). Figure 7 shows the bosonic momentum distribution from weak to strong BF coupling for different bosonic concentrations $x$ and zero bosonic repulsion, $\eta_B = 0$. A peculiar feature common to all panels is the appearance of a peak for small but non-zero momenta,

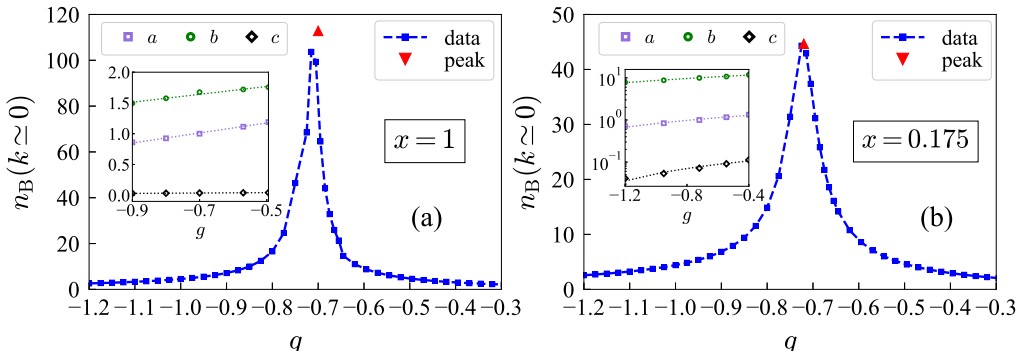

Figure 9: Bosonic momentum distribution $n_B(k)$ evaluated at small momentum $k \simeq 0.003k_F$ as a function of BF coupling $g$, at BB repulsion $\eta_B = 0$, for bosonic concentration (a) $x = 1$ and (b) $x = 0.175$. Triangle: peak value obtained from Eq. (53). Dashed line: guide to the eye. Insets: parameters $a, b, c$ of Eq. (50), where the dotted lines are linear fits.

depending on the concentration and coupling. In particular, for very weak coupling, the peak persists for all concentrations (full line in all panels) in contrast to higher couplings where it soon disappears. One is thus led to think that a weak-coupling mean-field effect may be at the origin of the peak, primarily due to the unpaired atoms dressing up the weakly interacting bosons, hence a sort of polaronic effect. However, for a definite identification of this peak, an analysis of the bosonic spectral weight function would be in order (this is postponed to future work).

At large $k$, the momentum distribution is expected to be dominated by short-range pairing correlations captured by the asymptotic expression (see Sec. 4.5)

$$n_B(k) \simeq \frac{\int \frac{d\mathbf{P}}{(2\pi)^2} \int \frac{d\Omega}{2\pi} T(\mathbf{P}, \Omega) e^{i\Omega 0^+}}{\left(\frac{k^2}{2m_r} - \mu_B - \mu_F\right)^2} \, . \tag{49}$$

The insets in Fig. 7 compare the numerical large-$k$ behavior of $n_B(\mathbf{k})$ with the corresponding asymptotic expression (49) (dotted lines).

In strict analogy to what is found in 3D [91], and preparing for the universality of the condensate fraction that will be illustrated in Sec. 4.4, it is instructive to inspect the boson momentum distribution divided by the boson concentration $x$. Figure 8 shows that the momentum distributions $n_B(\mathbf{k})$ corresponding to different values of $x$ collapse on top of each other once divided by the concentration $x$, except for a region of small $\mathbf{k}$. This universality, which occurs from weak to intermediate coupling ($g \lesssim 0$), suggests that in this coupling range the bosons can be treated as nearly independent of each other (but interacting with the medium), such that $n_B(\mathbf{k}) \simeq n_B n_{x\to 0}(\mathbf{k})$, with $n_{x\to 0}(\mathbf{k}) \equiv \lim_{x\to 0} n_B(\mathbf{k})/n_B$. We stress that this is not obvious a priori, even in the absence of a direct BB repulsion (as in the case we are considering here), since indirect interactions mediated by the fermions are possible. Indeed, this is what happens at small momenta $k$, as signaled by the deviation of $n_B(\mathbf{k})$ from such an independent particle picture in this momentum range.

An interesting feature is also observed when plotting the limiting value for vanishing momentum ($n_B(\mathbf{k} \to 0)$) of the momentum distribution of non-condensed bosons as a function of coupling, as shown in Fig. 9 for two representative values at large and small boson concentrations. By keeping $k/k_F$ fixed at a small value and scanning the BF interaction, a prominent peak appears close to $g \simeq -0.7$ which eventually diverges in the limit $k = 0$. We stress that in the present case, the BB repulsion is turned off and thus there is no anomalous self-energy

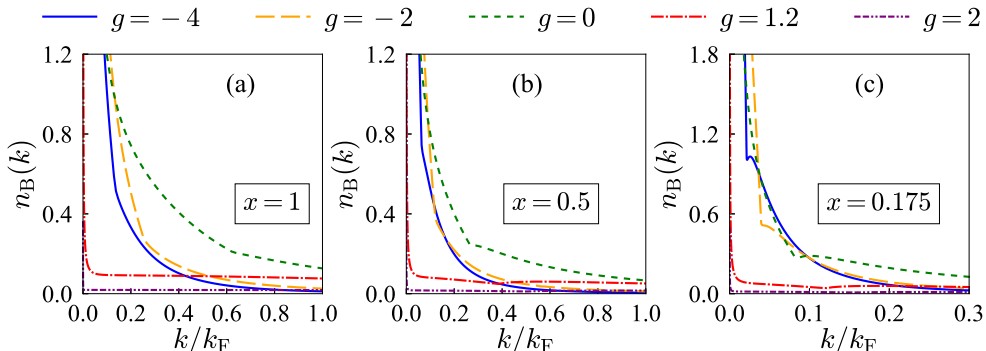

Figure 10: Bosonic momentum distribution $n_B(k)$ as a function of $k/k_F$, for different values of boson-fermion coupling $g$ at BB repulsion $\eta_B = 0.1$. Different cases of bosonic concentration $x$ are reported: (a) $x = 1$; (b) $x = 0.5$; (c) $x = 0.175$.

term (which would yield $n_B(\mathbf{k} \to 0) \to \infty$ for all values of $g$, as standard from Bogoliubov theory).

It is possible to elucidate the origin of this behavior by assuming a generic expansion for the boson self-energy at small $\mathbf{k}$ and $\omega$

$$\Sigma_B(\mathbf{k}, \omega) = \Sigma_B(\mathbf{0}, 0) + a\, i\omega + b\, \omega^2 + c\, k^2 \,, \tag{50}$$

with $a, b, c$, real. When inserted in Eq. (14) with $\Sigma_{12} = 0$, Eq. (50) yields

$$G_B(\mathbf{k}, \omega) \simeq \frac{1}{(1-a)\, i\omega - \frac{\mathbf{k}^2}{2m_B^*} - b\omega^2} \,, \tag{51}$$

where $1/m_B^* = 1/m_B + c$ and the Hugenholtz-Pines condition has been used. The coefficients $a, b, c$ are obtained by fitting the data for the bosonic self-energy at small $\omega$ and $\mathbf{k}$, for fixed values of $g$. Then, through a simple linear regression, $a, b, c$ can be obtained as a function of $g$, as shown in the insets of Fig. 9. One can notice that for $g \simeq -0.7$, $a = 1$ and $b > 0$, resulting in the following expression for the momentum distribution (16)

$$n_B(\mathbf{k}) \simeq \int_{-\omega_{IR}}^{+\omega_{IR}} \frac{d\omega}{2\pi} \frac{1}{\frac{\mathbf{k}^2}{2m_B^*} + b\omega^2} + 2 \int_{\omega_{IR}}^{+\infty} \frac{d\omega}{2\pi} \Re[G_B(\mathbf{k}, \omega)] \,, \tag{52}$$

where $\omega_{IR}$ is an appropriate infrared cutoff defining the range of validity of the above expansion. For small enough $\mathbf{k}$ the second integral is a subleading contribution and one obtains to leading order

$$n_B(\mathbf{k}) \approx \frac{1}{2} \sqrt{\frac{m_B^*}{m_F} \frac{1}{b E_F}} \frac{k_F}{k} \,, \qquad k \to 0 \,. \tag{53}$$

One sees in Fig. 9 that this asymptotic expression matches well the peak value of $n_B(k \simeq 0)$ as a function of $g$.

This short exercise reveals that at $g \simeq -0.7$ the energy spectrum at small momenta provided by the poles of the Green's function (51) changes from being quasi-particle like ($\omega \propto \pm k^2$ when $a \lessgtr 1$, respectively) to collective (phononic) mode type ($\omega^2 \propto k^2$ when $a = 1$). In contrast to the weakly repulsive Bose gas where the energy spectrum is given by $\omega^2 = c_s^2 k^2$ with $c_s$ the sound velocity, here the latter turns out to be purely imaginary ($\omega^2 = -k^2/(2m_B^* b)$) implying the mechanical instability of the Bose component. This is not unexpected since the indirect interaction among bosons mediated by the fermionic medium is attractive at any coupling [3] thus making the mixture mechanically unstable per se. However, we note that the

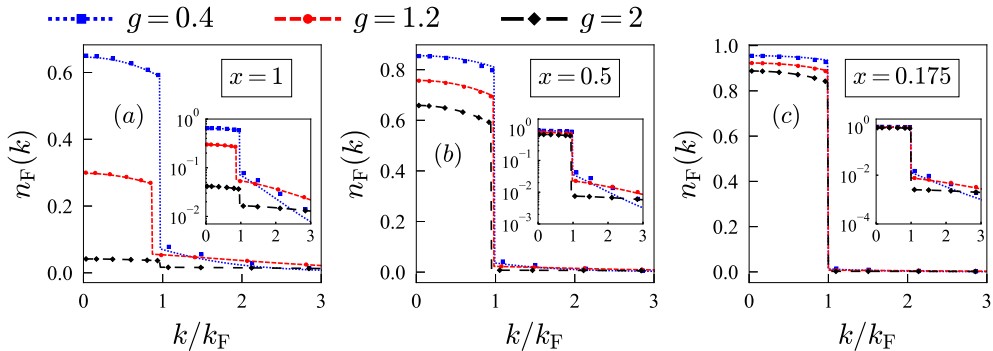

Figure 11: Fermionic momentum distribution $n_F(k)$ as a function of $k/k_F$ for bosonic concentrations (a) $x = 1$, (b) $x = 0.5$ and (c) $x = 0.175$, for different strong-coupling values of the BF coupling $g$ and BB repulsion $\eta_B = 0$. Symbols indicate numerical data, while lines correspond to the strong-coupling approximation (39) using the same thermodynamical parameters as in the numerical calculation. Insets: same quantities in log-scale.

microscopic mechanism of mechanical collapse is generally due to quasi-particle or incoherent excitations [84], except for $g \simeq -0.7$ where instead a phononic collective mode sets in.

For this reason, one needs to include a BB repulsion in to stabilize the system against the aforementioned effects. We thus consider the case where a direct BB repulsion is turned on, causing a divergence at zero $k$ to be present at all $g$'s, as expected within Bogoliubov theory. Calculations for a relatively strong boson-boson repulsion $\eta_B = 0.1$ are reported in Fig. 10. Clearly, the main differences with respect to the case of zero repulsion occur at small momenta, where the occupation of low momenta states predicted by Bogoliubov theory dominates. Regarding the large momentum regime (not reported in Fig. 10), it is well described by expression (61) discussed below.

Concerning the Fermi momentum distribution, we first discuss its comparison with the strong-coupling expression (39) with a two-fold aim: to untangle the different contributions from atomic and molecular fermions based on what we learned in Sec. 3.2 and, at the same time, to check to what extent the approximate expression (38) for $G_F(\mathbf{k}, \omega)$, from which Eq. (39) for $n_F(\mathbf{k})$ immediately results, provides a good approximation for $G_F(\mathbf{k}, \omega)$. Fig. 11 shows the numerical results for $n_F(\mathbf{k})$ as symbols and the expression (39) as lines for several values of BF coupling $g$ and (a) $x = 1$, (b) $x = 0.5$ (c) $x = 0.175$, using the same thermodynamic parameters for both calculations. The use of the same thermodynamic parameters allows for a separate check of the approximation (38) for $G_F(\mathbf{k}, \omega)$, independently of the remaining strong-coupling approximations (46) and (47) affecting the values of the thermodynamic parameters. The agreement is remarkably good for $g \gtrsim 1$, and remains reasonable even at the lowest coupling $g = 0.4$ considered in Fig. 11. This demonstrates that the fermionic Green's function is well captured by the "BCS-like" expression (38) in an extended coupling range and not only for large values of $g$, where it is expected to hold.

We also note that the discussion on paired and unpaired fermions of Secs. 3.2 and 3.4 applies here, with the tail of $n_F(\mathbf{k})$ representing the internal part of the molecular wave-function and the Fermi step being associated with unpaired yet correlated fermions. We notice that, when $g$ is reduced, discrepancies between the asymptotic curve and the numerical data occur first for the higher boson concentration, and they occur in the tail of $n_F(\mathbf{k})$. This is related to the onset of the continuum of particle-hole excitations becoming comparable to the molecular binding energy, thus making the BCS-like expression (39) less valid.



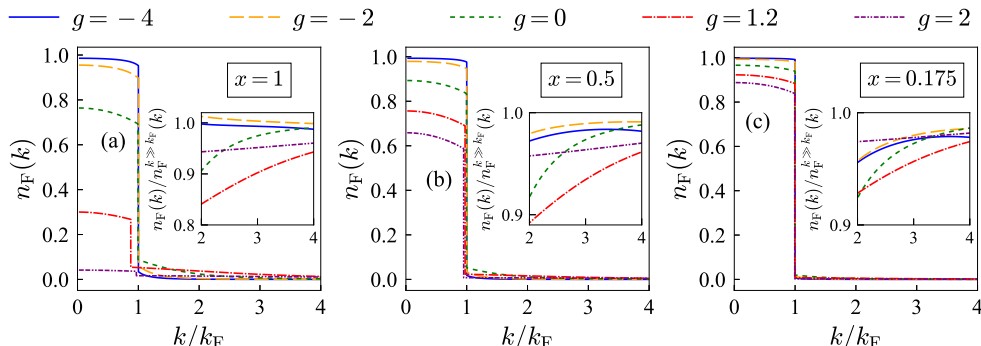

Figure 12: Fermionic momentum distribution $n_F(k)$ as a function of $k/k_F$, for different values of the BF coupling $g$ and BB repulsion $\eta_B = 0$. Different cases of bosonic concentration $x$ are reported: (a) $x = 1$, (b) $x = 0.5$, (c) $x = 0.175$. Insets: ratio of $n_F(k)$ to its large-momentum behavior given by Eq. (60).

For generic coupling strength, the numerical Fermi momentum distributions are shown in Fig. 12 at different bosonic concentrations $x$. We only report the case $\eta_B = 0$, since the effect of the BB repulsion is only limited to a slight renormalization of the Fermi step. We find a qualitatively analogous phenomenology to Fig. 11 even though the latter is obtained in the strong-coupling limit of the theory. This is because the effective interaction acting on fermions is given by an interplay between $g$ and $x$, that is, even if $g$ is strong, reducing $x$ results in an effectively weaker $g$. In addition, we observe that the position of the Fermi step does not change significantly from weak to strong-coupling, in contrast to the 3D case where the Fermi sphere is emptied by the vanishing of the Fermi wave vector [86]. In contrast, in 2D the destruction of the Fermi surface occurring with coupling or concentration takes place due to a strong renormalization of the fermionic quasi-particles, as described in Secs. 3.2 and 3.4, and it is their weight rather than their associated Fermi momentum that vanishes.

Finally, in the inset, the fermionic momentum distribution is compared to the square modulus of the composite fermion wavefunction (60) in the large momentum limit where their ratio is expected to approach one.

## 4.3 Boson and fermion chemical potentials

We now discuss the results obtained for the Bose and Fermi chemical potentials. In the weak-coupling regime, one can expand the $T$-matrix in the small parameter $1/g$ and obtain analytical expressions for both chemical potentials. These calculations are carried out in detail in [104] and we only report the final results here

$$\mu_B = \frac{4\pi n_0 \eta_B}{m_B} + \frac{E_F}{g}\left[1 - \frac{1}{g}\left(\ln 2 - \frac{1}{2}\right)\right], \tag{54}$$

$$\mu_F = E_F + x\frac{E_F}{g}\left[1 + \frac{1}{g}(1 - \ln 2)\right]. \tag{55}$$

We stress that the above perturbative expressions have been tested against independent fixed-node diffusion Quantum Monte Carlo calculations in [104]. The first term in Eq. (54) arises from the standard Bogoliubov theory in 2D [94], while the second contribution stems from the BF interaction. One sees in Fig. 13 that in the weak-coupling regime, the numerical curves for both chemical potentials neatly approach the asymptotic limits (54,55) (dotted line on the left side). For the bosonic chemical potential, this occurs in a universal way, i.e., independently of the value of $x$, as required by Eq. (54) which does not depend on the bosonic

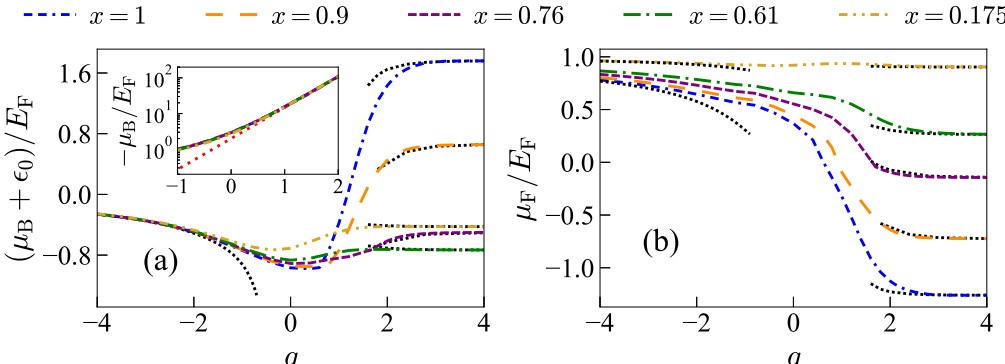

Figure 13: (a): Bosonic chemical potential $\mu_B$ (shifted by the binding energy $\varepsilon_0$), in units of $E_F$, as a function of the BF coupling $g$, for different values of bosonic concentration $x$, and BB repulsion $\eta_B = 0$. Dotted lines on the left: weak-coupling expression (54). Dotted lines on the right: strong-coupling approximation of Sec. 3. Inset: comparison between $-\mu_B$ and the binding energy $\varepsilon_0$ (dotted line), both in units of $E_F$, as a function of $g$. (b): Corresponding fermionic chemical potential $\mu_F$ in units of $E_F$. Dotted lines on the left: weak-coupling expression (55) (reported for clarity only for $x = 1$ and $x = 0.175$). Dotted lines on the right: strong-coupling approximation of Sec. 3.

concentration (and $\varepsilon_0$ is exponentially suppressed in this regime). In the opposite regime, both chemical potentials approach their strong-coupling benchmarks, depicted by the dotted lines on the right side of the graph, which are seen to be strongly dependent on concentration. Note that, for the boson chemical potential, Fig. 13(a) reports the quantity $\mu_B + \varepsilon_0$ to remove from the bosonic chemical potential the leading strong-coupling contribution (given by $-\varepsilon_0$). The inset avoids this subtraction and illustrates on a logarithmic scale how the bosonic chemical potential approaches the binding energy as $g$ increases, thus showing that all bosons are paired with fermions at large values of $g$.

In particular, we have verified that for $g \simeq 1$ the relative difference between $\mu_B$ and $-\varepsilon_0$ goes below 5% for all concentrations. At about the same coupling strength, the fermion chemical potential $\mu_F$ changes sign for the case $x = 1$ (while clearly $\mu_F$ is progressively less affected by interaction as $x$ decreases, since there are fewer bosons to interact with). We thus identify the value $g \simeq 1$ as the border of the region in which almost all bosons are paired with atomic fermions into composite fermions (which in general remain hybridized with unpaired fermions and residual condensed bosons, except for the case $x = 1$).

Finally, Fig. 14 shows that at finite $\eta_B = 0.1$ the results remain qualitatively the same, with quantitative changes that affect only $\mu_B$, as also expected from the weak-coupling expression (54). The fermionic chemical potential is instead unaffected, in line with Eq. (55). On the other hand, in the strong-coupling regime, the boson-boson repulsion is essentially irrelevant due to the dominant dimer-atom repulsion. In summary, our results for the chemical potentials show that the present approach is able to recover both the pairing of bosons with fermions into composite fermions in the strong-coupling limit and the perturbative results of Ref. [104] in the opposite weak-coupling limit.

## 4.4 Boson condensate density and atom-molecule hybridization energy scale

We now discuss the results for the bosonic condensate density $n_0$ and the related hybridization energy scale $\Delta_0$. Figure 15 reports the value of the condensate fraction as a function of coupling (for $\eta_B = 0$ in panel (a) and $\eta_B = 0.1$ in panel (c)). A striking feature of Fig. 15 is

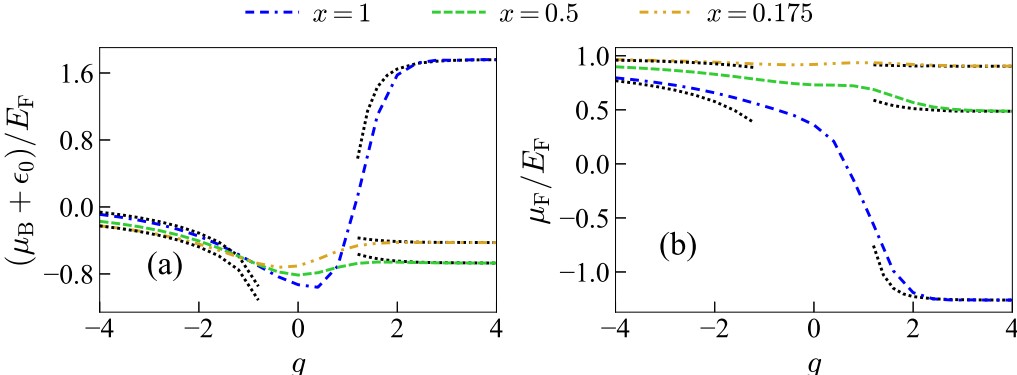

Figure 14: (a): Bosonic chemical potential $\mu_B$ (shifted by the binding energy $\varepsilon_0$), in units of $E_F$, as a function of the BF coupling $g$, for different values of bosonic concentration $x$, and BB repulsion $\eta_B = 0.1$. Dotted lines on the left: weak-coupling expression (54) (reported for clarity only for $x = 1$ and $x = 0.175$). Dotted lines on the right: strong-coupling approximation of Sec. 3. (b): Corresponding fermionic chemical potential $\mu_F$ in units of $E_F$. Dotted lines on the left: weak-coupling expression (55) (reported for clarity only for $x = 1$ and $x = 0.175$). Dotted lines on the right: strong-coupling approximation of Sec. 3.

the nearly universal behavior of $n_0$ with respect to concentration, in complete analogy to what was found in 3D [91] and recently confirmed experimentally in [33].

Figure 15(a) also reports the results for the polaron quasiparticle residue $Z$ previously obtained with Diagrammatic Monte Carlo simulations (circles) [99] and with the non-self-consistent $T$-matrix approach [123] (squares). This is because in 3D it was found [33, 91] that the universal condensate fraction essentially matches the quasiparticle residue $Z$ of the Fermi polaron (which in our Bose-Fermi mixture corresponds to the limit of a single boson immersed in a Fermi gas).

Such identification between the universal condensate fraction and $Z$ is challenged by the present results in 2D. One sees that, even though the condensate fraction $n_0/n_B$ and the polaron quasiparticle residue $Z$ share the same qualitative behavior, significant quantitative discrepancies between the two quantities are present. We emphasize that the existence of such a discrepancy cannot be ascribed to the specific choice of the self-energy used in our calculation. One sees indeed that such a discrepancy occurs also when $Z$ is obtained with the non-self-consistent $T$-matrix approach (squares in Fig. 15(a)), which is exactly the same approach used here when carried over to the single-impurity limit. As an independent check, we have also calculated $Z$ from the derivative of the bosonic self-energy with respect to the frequency by using the same code used at finite $x$, adapted to the single-impurity limit. Our results (diamonds) fully agree with the corresponding results for $Z$ from [123], thus showing that the discrepancy between $Z$ and the condensate fraction cannot be ascribed to possible numerical errors. We therefore conclude that the equivalence between the quasiparticle residue $Z$ of the Fermi polaron and the universal condensate fraction observed in 3D is only approximate, and the differences between these two quantities are amplified in 2D.

In this respect, reconsidering the 3D case and focusing on unitarity, by comparing the data from [91] for $n_0/n_B$ at the lowest concentration ($x = 0.175$) with the polaron residue obtained with the non-self-consistent T-matrix approach [124] ($Z^{TMA}$) or with the diagrammatic Monte-Carlo method [124] ($Z^{dMC}$), one has $n_0/n_B = 0.74$, $Z_{pol}^{TMA} = 0.80$, $Z^{dMC} = 0.76$ for $m_B/m_F = 0.575$; $n_0/n_B = 0.73$, $Z^{TMA} = 0.78$, $Z^{dMC} = 0.76$ for $m_B/m_F = 1$; $n_0/n_B = 0.60$, $Z^{TMA} = 0.67$, $Z^{dMC} = 0.65$ for $m_B/m_F = 5$. One sees that $n_0/n_B$ and $Z^{TMA}$ differ by ap-

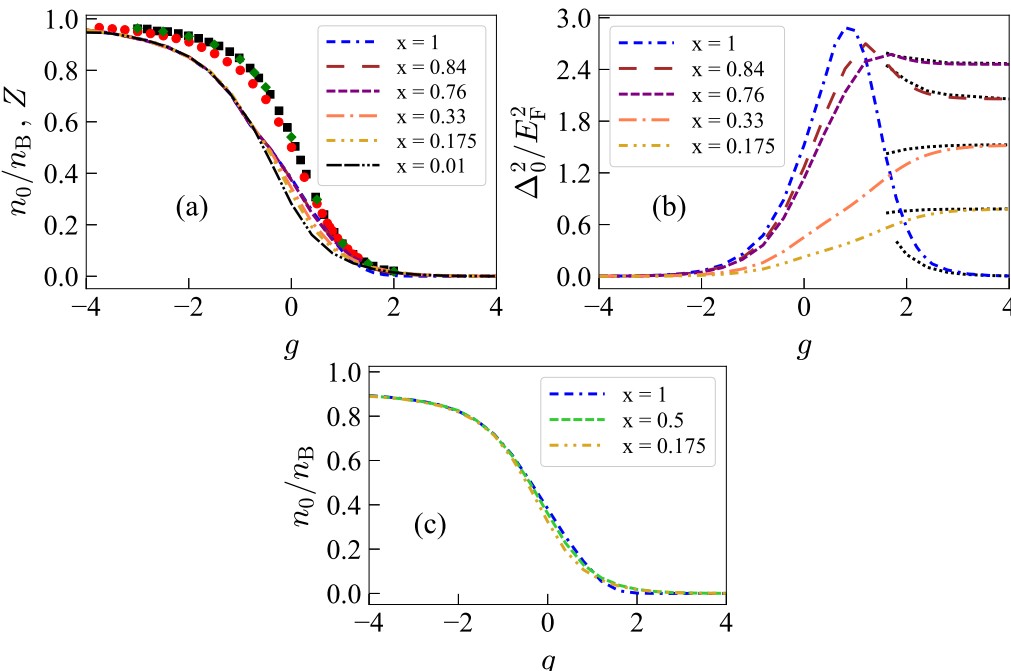

Figure 15: (a): Bosonic condensate fraction $n_0/n_B$ as a function of the BF coupling parameter $g$, for different values of boson concentration $x$ and BB repulsion $\eta_B = 0$. Circles: Diagrammatic Monte Carlo results for the polaron quasiparticle residue $Z$ [99]. Squares: non-self-consistent $T$-matrix approximation results for the polaron quasiparticle residue $Z$ [123]. Diamonds: non-self-consistent $T$-matrix approximation results for the polaron quasiparticle residue $Z$ by us. (b): Square of hybridization energy scale $\Delta_0^2 = 2\pi\varepsilon_0 n_0/m_r$ (in units of $E_F^2$), as a function of the BF coupling $g$, for different values of the bosonic concentration $x$ and boson-boson repulsion $\eta_B = 0$. Strong-coupling benchmarks from Sec. 3 are also reported as dotted lines on the right. (c): Bosonic condensate fraction $n_0/n_B$ as a function of the BF coupling parameter $g$, for different values of boson concentration $x$ and BB repulsion $\eta_B = 0.1$.

proximately 6% for equal masses, and the difference increases by changing the mass ratio, exceeding 10% for $m_B/m_F = 5$. (The difference between $n_0/n_B$ and $Z^{dMC}$ is slightly smaller, but the comparison with $Z^{TMA}$ is more meaningful, since $Z^{TMA}$ and $n_0/n_B$ are calculated within the same approximation.) This further indicates that the equivalence observed in 3D between the polaron residue and $n_0/n_B$ is only approximate; the present calculation indicates that this approximate degeneracy is lifted in 2D.

In fact, and contrary to what was argued in [91], there is no reason why the limit for $x \to 0$ of the condensate fraction should coincide with the polaron residue $Z$. The first quantity is defined by the ratio between the number of condensed bosons $N_0$ and the total number of bosons $N_B$ in the thermodynamic limit ($N_B \to \infty$, $V \to \infty$, at fixed $n_B$ and $x$, $V$ being the volume). Only in this limit is a condensate fraction well defined. So, no matter how small $x$ is, the number of bosons $N_B$ will always scale to infinity in the thermodynamic limit, while in the polaron limit one has instead $N_B = 1$. Therefore, the occupancy of the zero-momentum state for a single impurity cannot be in general related to the ratio $N_0/N_B$ in the limit $x \to 0$. It is a matter of the order of limits. In the first case (condensate fraction), one first fixes $x$, takes the thermodynamic limit $V \to \infty$, and finally lets $x \to 0$. In the second case (polaron $Z$), one first fixes $N_B = 1$ and then takes the limit $V \to \infty$ in such a way that $x = 1/(n_F V) \to 0$. Hence, in

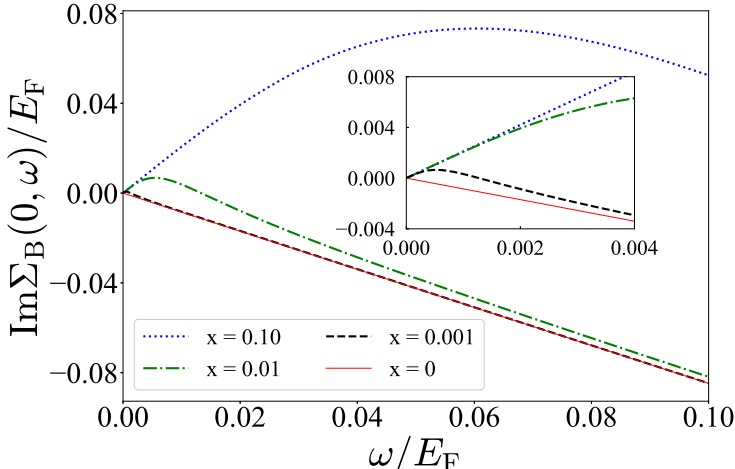

Figure 16: Imaginary part of the bosonic self-energy, in units of $E_F$, as a function of the frequency $\omega$ along the imaginary axis for different values of boson concentration $x$ at BF coupling $g = 0$ and BB repulsion $\eta_B = 0$. The case $x = 0$ corresponds to the calculation in the polaron limit of a single impurity in a Fermi bath.

this second case the thermodynamic limit and the limit $x \to 0$ are taken simultaneously, while in the first case the thermodynamic limit is taken first, followed by the limit $x \to 0$. It is thus clear that the two limits are different:

$$\lim_{x \to 0} \lim_{V \to \infty} \frac{n_B(k=0; N_B = x N_F; N_F/V = \text{const})}{N_B} \neq \lim_{V \to \infty} \frac{n_B(k=0; N_B = 1; N_F/V = \text{const})}{1} . \quad (56)$$

A interesting related question is how the polaron residue $Z$ is obtained from the limiting behavior of the bosonic self-energy when $x \to 0$. One has [117, 125]

$$Z = \lim_{\omega' \to 0} \frac{1}{\left| 1 - \frac{\partial \text{Re} \Sigma_p^R(0,\omega')}{\partial \omega'} \right|} = \lim_{\omega \to 0^+} \frac{1}{\left| 1 - \frac{\partial \text{Im} \Sigma_p(0,\omega)}{\partial \omega} \right|} . \quad (57)$$

Here, $\Sigma_p^R(0,\omega')$ is the retarded polaron self-energy calculated at zero momentum and at the real frequency $\omega'$, while $\Sigma_p(0,\omega)$ is its analytic extension into the upper complex plane calculated at the imaginary frequency $i\omega$. For a given BF coupling strength, one has $\Sigma_p(0,\omega) = \lim_{x \to 0} \Sigma_B(0,\omega;x)$, where $\Sigma_B(0,\omega;x)$ is the boson self-energy at finite $x$ calculated at the imaginary frequency $i\omega$, for $\eta_B = 0$. Hence,

$$Z = \lim_{\omega \to 0^+} \lim_{x \to 0} \frac{1}{\left| 1 - \frac{\partial \text{Im} \Sigma_B(0,\omega;x)}{\partial \omega} \right|} . \quad (58)$$

One may also consider, at finite $x$, the quasiparticle residue of the bosons at vanishig momentum and frequency:

$$Z(x) = \lim_{\omega \to 0^+} \frac{1}{\left| 1 - \frac{\partial \text{Im} \Sigma_B(0,\omega;x)}{\partial \omega} \right|} . \quad (59)$$

It is evident that in general $Z$ might be different from $\lim_{x \to 0} Z(x)$ due to the different order in the two limits involved. That this is indeed the case, it is suggested by the results of Sec. 4.2 (in particular the discussion of Fig. 9), which show that $Z(x) \gg 1$ around $g = -0.7$, against a corresponding value of $Z \simeq 0.9$.

To further analyze this behavior at a more generic BF coupling, Fig. 16 reports $\text{Im}\Sigma_{\text{B}}(0,\omega)$ at $g=0$ for three different concentrations approaching the limit $x \to 0$, together with its expected limiting value for $x=0$, $\text{Im}\Sigma_{\text{p}}(0,\omega)$. One sees that $\text{Im}\Sigma_{\text{B}}(0,\omega)$ tends indeed to $\text{Im}\Sigma_{\text{p}}(0,\omega)$. However, the slope of $\text{Im}\Sigma_{\text{B}}(0,\omega)$ in $\omega=0$ is nearly independent of $x$ and, for $x \to 0$, tends to a value that is completely different from the slope of $\text{Im}\Sigma_{\text{p}}(0,\omega)$ in $\omega=0$. In this way, one has $\lim_{x\to 0} Z(x) = 0.93$, against a value of $Z = 0.54$.

Returning to Fig. 15, another difference is observed with respect to the 3D case in the behavior of $n_0$ for strong coupling. While in 3D $n_0$ drops identically to zero for values of $g$ larger than a critical coupling $g_c$, in 2D $n_0$ never vanishes (even though it becomes exponentially small with the coupling $g$). This in turn implies that the hybridization energy scale $\Delta_0 \propto (n_0 \varepsilon_0)^{1/2}$ remains in general finite even at large values of $g$, except for the case $x=1$ (besides the trivial case $x=0$), as shown in Fig. 15(b). The absence of a critical coupling in 2D here obtained recalls the absence of the polaron-to-molecule transition, which, as mentioned in Sec. 2.2, is a known shortcoming of the present approach when applied to the single impurity problem in 2D [97,98]. This defect originates from an overestimate of the atom-molecule repulsion in the strong-coupling limit of the theory [97]. This overestimate persists also at finite $x$ (more on this point in Sec. 5). We cannot therefore exclude that the absence of a critical coupling strength in 2D is just an artifact of the present approximation, in particular at small values of $x$, which are closer to the single impurity limit.

## 4.5 Tan's contact parameter and composite-fermion density

For large momenta, the fermionic and bosonic distribution functions have the following asymptotic behaviors

$$n_{\text{F}}^{k \gg k_{\text{F}}}(k) \simeq \frac{C_{\text{BF}}/4m_r^2}{\left(\frac{k^2}{2m_r} - \mu_{\text{B}} - \mu_{\text{F}}\right)^2}, \tag{60}$$

$$n_{\text{B}}^{k \gg k_{\text{F}}}(k) \simeq \frac{C_{\text{BF}}/4m_r^2}{\left(\frac{k^2}{2m_r} - \mu_{\text{B}} - \mu_{\text{F}}\right)^2} + \frac{\Sigma_{12}^2}{4\left(\frac{k^2}{2m_{\text{B}}} - \mu_{\text{B}}\right)^2}, \tag{61}$$

whose leading order term, proportional to $k^{-4}$, defines the Tan's contact parameter $C_s = n_s(k)k^4$, $s = \text{F},\text{B}$, for fermions and bosons respectively [118–120]. The coefficient $C_{\text{BF}}$ gives the contribution to Tan's contacts due to BF pairing. Within our $T$-matrix self-energy approach, it can be shown to be given by

$$C_{\text{BF}} = 4m_r^2 \int \frac{d\mathbf{P}}{(2\pi)^2} \int \frac{d\Omega}{2\pi} T(\mathbf{P},\Omega)e^{i\Omega 0^+}, \tag{62}$$

similarly to what one obtains for analogous $T$-matrix approaches in the BCS-BEC crossover problem [126,127].

We show the results for $C_{\text{BF}}$ of Eq. (62) in Fig. 17(a) from weak to strong BF attraction for different values of the bosonic concentration. The same quantity once divided by the boson density, $C_{\text{BF}}/n_{\text{B}}$, displays a universal behavior (see inset), in line with what already discussed for the condensate fraction (sec. 4.4) and bosonic momentum distribution (sec. 4.2). For this quantity, one sees that universality occurs for all couplings.

In the strong-coupling regime, $C_{\text{BF}}$ is seen in Fig. 17(a) to diverge exponentially with coupling. Specifically, as shown in the inset, the dimensionless quantity $C_{\text{BF}}/(m_{\text{F}}^2 E_{\text{F}}^2)$ scales like $2\varepsilon_0/E_{\text{F}}$ (dotted line in the inset), with the binding energy $\varepsilon_0/E_{\text{F}} = 2e^{2g}$. In this limit, once

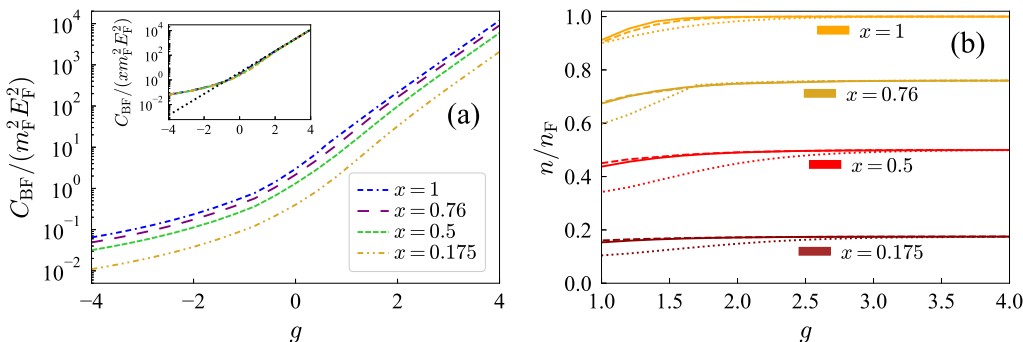

Figure 17: (a): Dimensionless Tan's contact constant $C_{\mathrm{BF}}/m_{\mathrm{F}}^2 E_{\mathrm{F}}^2$ as a function of the BF coupling $g$, for different values of the bosonic concentration $x$ and BB repulsion $\eta_{\mathrm{B}} = 0$. Inset: same data divided by the boson concentration $x$, compared with $2\varepsilon_0/E_{\mathrm{F}}$ (dotted line). (b): Composite-fermion number density $n_{\mathrm{CF}}$ in units of $n_{\mathrm{F}}$ (full lines), as obtained from Eq. (33) for different values of the bosonic concentration ($x = 1.0, 0.76, 0.5, 0.175$, from top to bottom) compared with $C_{\mathrm{BF}}/(8\pi m_r \varepsilon_0 n_{\mathrm{F}})$ (dotted lines) and $(n_{\mathrm{B}} - n_0)/n_{\mathrm{F}}$ (dashed lines).

$T(\mathbf{P}, \Omega)$ is replaced in Eq. (62) by $T_{\mathrm{SC}}(\mathbf{P}, \Omega)$, one has indeed

$$C_{\mathrm{BF}} \approx 4m_r^2 \int \frac{d\mathbf{P}}{(2\pi)^2} \int \frac{d\Omega}{2\pi} T_{\mathrm{SC}}(\mathbf{P}, \Omega) e^{i\Omega 0^+} \tag{63}$$

$$= 8\pi m_r \varepsilon_0 \int \frac{d\mathbf{P}}{(2\pi)^2} \int \frac{d\Omega}{2\pi} G_{\mathrm{CF}}(\mathbf{P}, \Omega) e^{i\Omega 0^+} \tag{64}$$

$$= 8\pi m_r \varepsilon_0 n_{\mathrm{CF}}, \tag{65}$$

thus showing that in the strong-coupling limit $C_{\mathrm{BF}}$ becomes proportional to the binding energy $\varepsilon_0$ multiplied by the density of BF pairs $n_{\mathrm{CF}}$ (where $n_{\mathrm{CF}}$ is defined by the strong-coupling expression (33)).

The presence of $n_{\mathrm{CF}}$ in Eq. (63) suggests comparing $n_{\mathrm{CF}}$ defined by Eq. (33) to the expression $C_{\mathrm{BF}}/(8\pi m_r \varepsilon_0 n_{\mathrm{F}})$, in order to see when deviations from the asymptotic expression occur. This is done in Fig. 17(b) for several values of the concentration $x$ (full lines and dotted lines, respectively). The deviations are most significant at small concentrations, but for $g \gtrsim 2.5$ the contact essentially coincides with its asymptotic expression for all cases. We also notice that for $g \gtrsim 1$, the quantity $n_{\mathrm{CF}}$ essentially coincides with the number of bosons out of the condensate, $n_{\mathrm{B}} - n_0$, (represented by the dashed lines in Fig. 17(b)). This justifies the assumption $n_{\mathrm{CF}} \simeq n_{\mathrm{B}} - n_0$ which we made in Sec. 3.4 when discussing the strong-coupling limit of our theory.

## 4.6 Identification of different physical regions in the coupling-concentration plane

Based on the results obtained in Secs. 3.2, 3.4, 4.3, and 4.4, the coupling ($g = -\ln(k_{\mathrm{F}} a_{\mathrm{BF}})$) vs. boson concentration ($x = n_{\mathrm{B}}/n_{\mathrm{F}}$) plane of a BF mixture can be divided into different physical regions. Specifically, in Sec. 3.2 and 3.4 two different regimes were identified when varying the bosonic concentration for a fixed strong BF attraction. Similarly, two different regimes were identified in Sec. 4.3 when varying the coupling strength, while the analysis of the condensate fraction of Sec. 4.4 added further insight into the evolution with the coupling strength.

Gathering all this information together, a schematic picture can be constructed as in Fig. 18, whereby a number of different physical regimes are detected:

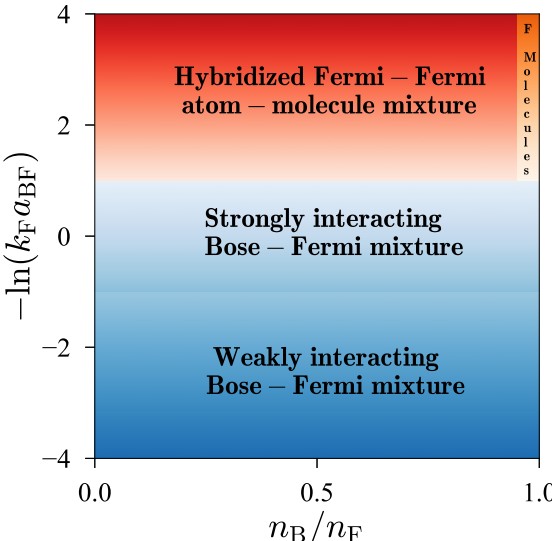

Figure 18: Schematic subdivision of the coupling vs. boson concentration plane of an attractive Bose-Fermi mixture in different physical regions.

- *Weakly interacting BF mixture*: a majority of weakly perturbed Fermi atoms coexist with a mildly depleted bosonic condensate; bosons out of the condensate marginally engage in molecule formation;

- *Strongly interacting BF mixture*: Fermi atoms are strongly paired with bosonic atoms, resulting in a significant depletion of the condensate, which in turn mediates a strong hybridization of molecular BF states with unpaired atomic ones;

- *Hybridized Fermi-Fermi mixture*: the condensate is almost fully depleted and bosons are fully paired up in molecules; a residual condensate density supports the hybridization of unpaired fermionic states with molecular states;

- *Non-interacting molecular gas*: Fermi molecules are the majority species; the hybridizing effect of the residual condensate vanishes, leaving a gas of non-interacting pure molecules.

## 5 Conclusions and outlook

In this work, we have analyzed the competition between boson condensation and BF pairing in 2D Bose-Fermi mixtures with an attractive and tunable BF interaction, extending previous work for the 3D case. We focused on the case with concentration of bosons $x = n_F/n_B \leq 1$, allowing for a full competition between pairing and condensation.

We have described analogies and differences with respect to the 3D case. Specifically, we have found that, like in 3D, the condensate is progressively depleted as the BF attraction increases and molecular correlations set in. However, in 2D the condensate never vanishes, but rather becomes exponentially small for large BF attractions. This marks a difference with respect to 3D, where the condensate was found to vanish beyond a critical coupling strength.

Similarly to the 3D case, we uncovered a nearly universal behavior for the condensate fraction and bosonic momentum distribution with respect to the boson concentration. However, in contrast to the 3D case, we have found that the universal condensate fraction does

not match the quasiparticle residue $Z$ of the Fermi polaron. It would be interesting to understand whether the equivalence between these two quantities observed in 3D has an intrinsic explanation or is simply an accidental degeneracy that is removed in 2D.

A further difference with respect to the 3D case is the hybrid nature of the composite fermions that form for sufficiently strong BF attraction. Rather unusual features are found within the momentum distribution function, except for the case $x = 1$ in which the mixture "fermionizes" completely in a gas of non-interacting point-like fermionic molecules, similarly to what is obtained in 3D for all $x \leq 1$. The origin of this difference for $x < 1$ is in the residual condensate density $n_0$ that, although small, proves sufficient to generate a significant hybridization through the hybridization energy-scale $\Delta_0 \propto (n_0 \varepsilon_0)^{1/2}$.

Finally, a discussion on possible refinements of the present approach is in order. The absence of a quantum phase transition for the condensate fraction found in the present work recalls the absence of the polaron-to-molecule transition that is found in the single impurity limit of a Fermi polaron when the polaron is described by a non-self-consistent $T$-matrix self-energy [97]. In this limit, it has been pointed out that a correct description of the atom-dimer interaction is crucial in bringing about this transition [97, 98].

In Sec. 3.1 and Sec. 3.2, we mentioned that the non-self-consistent $T$-matrix approximation takes into account the atom-dimer interaction at the level of the Born approximation, resulting in coupling-independent mean-field shifts $\Sigma^0_{\text{CF}}$ and $\Sigma^0_{\text{F}}$. While in 3D the Born approximation is qualitatively correct, in 2D these Hartree-like contributions definitely overestimate the effective repulsion between molecules and unpaired fermionic atoms, making the molecular state energetically less convenient, and favoring a state in which the molecule hybridizes with an unpaired state made of a fermionic atom and a boson belonging to the condensate. In order to go beyond the Born approximation, one should, in principle, include in a many-body theory the same kind of diagrams (involving two fermions repeatedly exchanging a boson) describing the atom-dimer scattering in the three-body problem [110, 128, 129] (see also [130]), a task that is far from trivial.

Insight from the polaron limit may suggest that the absence of a critical coupling strength in 2D is simply an artifact of the approximation for the self-energy here adopted. We do not exclude such a possibility, in particular at small values of $x$ which are closer to the single impurity limit. We, however, stress that, in experiments, the exponential vanishing of the condensate at large coupling strength, which we have found in the present work, would hardly be distinguishable from an exact vanishing of the condensate (leaving aside the fact that at finite temperature, as experimentally relevant, one should consider a quasi-condensate in 2D, introducing a further source of smearing of the transition).

Further on this point, recent work [90] has illustrated the relevance of three-body correlations in addressing the polaron-molecule quantum phase transition via an effective field theory approach based on a gradient expansion of the $T$-matrix for small momenta and frequencies and neglecting the condensation of bosons. In the present approach, we consider the full dependence of the $T$-matrix on momenta and frequencies in the presence of a Bose condensate, thus being able to span the full range of concentrations from a single impurity to equal densities. In particular, in the case $n_{\text{B}} = n_{\text{F}}$, three-body processes are expected to be negligible, making our results less sensitive to these diagrammatic contributions.

In addition, we point out that, since the polaron-to-molecule quantum phase transition is proven to occur specifically in the impurity limit, one cannot exclude that in the high-concentration regime, where the polaron picture breaks down, this transition could be suppressed or significantly modified by quantum fluctuations. It is known that in 3D, with a finite density of impurities, thermal and quantum fluctuations smoothen the discontinuity arising in the single impurity limit thus shifting the critical coupling to higher values [33, 131]. An analogous and stronger role may be played by quantum fluctuations in 2D. To settle this point,

a detailed calculation of the three-body diagrammatic contributions for finite densities beyond the polaronic regime should be addressed in future work.

It is also worth discussing the relevance of the present investigation to BF mixtures realized with transition metal dichalcogenides. The first issue is whether the effective interaction between excitons and excess charge carriers (the bosons and fermions of the mixture, respectively) can be modeled by a contact potential of the type utilized in the present work. In TMD systems, excitons and charge carriers can bind into a molecular trion state, which controls the effective BF interaction. The binding energy of the trion is typically at least one order of magnitude smaller than the exciton binding energy [132], and of the order of the Fermi energy of the excess charge carrier. Under these conditions, it has been shown that the effective BF interaction can indeed be reasonably modeled by a contact interaction [133]. A difference with respect to the model discussed in the present paper may instead be represented by the presence of Coulomb interactions between fermions, in particular intra-species interactions between excess charge carriers. However, it can be argued that [134], provided the doping is not too low [56, 90] (in order to be away from the regime of Wigner crystallization and reduced screening), their effect will mostly be to renormalize the single-particle dispersions of excess charges, an effect that can be safely described by the parameters of Fermi liquid theory (notably, by the effective mass of the charges). Concerning, finally, exciton-exciton interaction, it is expected to be small because of the tightly-bound nature of the excitons in TMDs that makes their dipole moment small, with a sign of the interaction that depends on the orientation of the dipoles. We have already mentioned that the mechanical stability of BF mixtures requires a repulsive interaction. In this respect, bilayer TMDs with interlayer excitons would be preferable, since in this case the dipole moments will be essentially parallel to each other (we notice that the distance between layers [42] is comparable with the exciton radius [90]), thus ensuring that the interaction is repulsive. However, it is not clear if such a repulsion would be sufficient to guarantee stability.

As a final remark, we wish to comment on possible extensions of the present work to finite temperature. In this case, the $T = 0$ condensed phase with long-range order will be replaced by a BKT superfluid phase with quasi-long-range order for the bosonic component. Similarly to $T = 0$, we expect BF pairing to compete with the bosonic superfluid phase. On a technical side, the main difficulty will be how to effectively include the BKT physics of the bosonic component in our diagrammatic scheme.

All the numerical data necessary to reproduce Figs. 3-17 are available online [135].

# Acknowledgments

We thank Dr. Andrea Guidini for contributing in an early stage of this work. We acknowledge the use of the parallel computing cluster of the Open Physics Hub at the Department of Physics and Astronomy of the University of Bologna.

**Funding information**  L.P. and P.P. acknowledge financial support from the Italian Ministry of University and Research (MUR) under project PRIN2022, Contract No. 2022523NA7. P.P. also acknowledges financial support from MUR project PE0000023-NQSTI (Italy) financed by the European Union - Next Generation EU and from European Union - NextGeneration EU through MUR under PNRR - M4C2 - I1.4 Contract No. CN00000013.

# A  Particle-particle ladder $\Gamma(\mathbf{P}, \Omega)$

We start from Eq. (5) for $\Gamma(\mathbf{P}, \Omega)$. For $\mu_{\mathrm{B}} \leq 0$, it corresponds to

$$\Gamma(\mathbf{P}, \Omega)^{-1} = \frac{1}{v_0^{\mathrm{BF}}} + \int \frac{d\mathbf{k}}{(2\pi)^2} \frac{1 - \Theta\left(-\xi_{\mathbf{P}-\mathbf{k}}^{\mathrm{F}}\right)}{\xi_{\mathbf{P}-\mathbf{k}}^{\mathrm{F}} + \xi_{\mathbf{k}}^{\mathrm{B}} - i\Omega} \tag{A.1}$$

$$= T_2(\mathbf{P}, \Omega)^{-1} - I_{\mathrm{F}}(\mathbf{P}, \Omega), \tag{A.2}$$

where $T_2$ is the off-shell two-body $T$-matrix in vacuum [104] which, for $\Omega \neq 0$, is given by

$$T_2(\mathbf{P}, \Omega)^{-1} = \frac{1}{v_0^{\mathrm{BF}}} + \int \frac{d\mathbf{k}}{(2\pi)^2} \frac{1}{\xi_{\mathbf{P}-\mathbf{k}}^{\mathrm{F}} + \xi_{\mathbf{k}}^{\mathrm{B}} - i\Omega} \tag{A.3}$$

$$= -\frac{m_r}{2\pi} \ln\left(\frac{\frac{P^2}{2M} - \mu_{\mathrm{F}} - \mu_{\mathrm{B}} - i\Omega}{\varepsilon_0}\right). \tag{A.4}$$

Note that Eq. (A.4) is obtained from Eq. (A.3) by expressing $v_0^{\mathrm{BF}}$ in terms of the boson-fermion binding energy $\varepsilon_0$ of the two-body problem in vacuum as in Eq. (2), regularizing in this way the ultraviolet divergence in Eq. (A.3).

The contribution $I_{\mathrm{F}}(\mathbf{P}, \Omega)$ stems from the presence of a degenerate fermionic component

$$I_{\mathrm{F}}(\mathbf{P}, \Omega) = \int \frac{d\mathbf{k}}{(2\pi)^2} \frac{\Theta\left(-\xi_{\mathbf{P}-\mathbf{k}}^{\mathrm{F}}\right)}{\xi_{\mathbf{P}-\mathbf{k}}^{\mathrm{F}} + \xi_{\mathbf{k}}^{\mathrm{B}} - i\Omega}, \tag{A.5}$$

therefore it is absent if $\mu_{\mathrm{F}} \leq 0$.

For $\mu_{\mathrm{F}} > 0$ and $\Omega \neq 0$, Eq. (A.5) can be expressed in closed form as

$$I_{\mathrm{F}}(\mathbf{P}, \Omega) = \frac{m_r}{2\pi} \left[ \ln(\mathcal{A}) - \ln\left(\frac{P^2}{2M} - \mu_{\mathrm{F}} - \mu_{\mathrm{B}} - i\Omega\right) - i\pi \mathrm{sgn}(\Omega) \right], \tag{A.6}$$

where

$$\mathcal{A} = \frac{z}{2} + \frac{P^2}{2M\gamma_m} + \mathrm{sgn}(\mathrm{Re}[z]) \sqrt{\left(\frac{z}{2}\right)^2 - \frac{P^2}{2m_{\mathrm{B}}\gamma_m} \mu_{\mathrm{F}}}, \tag{A.7}$$

and

$$z = \mu_{\mathrm{B}} - \frac{\mu_{\mathrm{F}}}{\gamma_m} - \frac{P^2}{2m_{\mathrm{B}}} + i\Omega, \qquad \gamma_m = \frac{m_{\mathrm{B}}}{m_{\mathrm{F}}}. \tag{A.8}$$

Note that, when inserted in Eq. (A.2), the second term in Eq. (A.6) perfectly cancels the logarithm of the numerator of Eq. (A.4), yielding

$$\Gamma(\mathbf{P}, \Omega) = \frac{2\pi/m_r}{\ln(\varepsilon_0/\mathcal{A}) + i\pi \mathrm{sgn}(\Omega)}. \tag{A.9}$$

For equal masses and real frequencies, an equation of the same form as Eq. (A.9) was first obtained in [123].

Finally, we derive strong-coupling expressions for $T_2(\mathbf{P}, \Omega)$ and $I_{\mathrm{F}}(\mathbf{P}, \Omega)$. For $T_2(\mathbf{P}, \Omega)$, we first write $\mu_{\mathrm{F}} + \mu_{\mathrm{B}} = \mu_{\mathrm{CF}} - \varepsilon_0$ in Eq. (A.2) and obtain

$$T_2(\mathbf{P}, \Omega)^{-1} = -\frac{m_r}{2\pi} \ln\left(1 + \frac{\frac{P^2}{2M} - \mu_{\mathrm{CF}} - i\Omega}{\varepsilon_0}\right) \tag{A.10}$$

$$\approx \frac{m_r}{2\pi\varepsilon_0} \left(i\Omega - \frac{P^2}{2M} + \mu_{\mathrm{CF}}\right), \tag{A.11}$$

Figure 19: Diagrammatic representation of Eq. (B.1) for the 3-body $T$-matrix $T_3$. The double line with an arrow or single line with an arrow represents a $T_2$ or a vacuum fermionic (F) or bosonic (B) propagator, respectively. Lines without arrows are dangling connection points defining the momentum flow.

where we have assumed $\Omega, P^2/2M, \mu_{\text{CF}} \ll \varepsilon_0$ in the strong-coupling limit $\varepsilon_0 \to \infty$.

For $I_\text{F}(\mathbf{P}, \Omega)$, after a shift of the integration variable and introducing again $\mu_{\text{CF}}$, we write

$$I_\text{F}(\mathbf{P}, \Omega) = \int \frac{d\mathbf{k}}{(2\pi)^2} \frac{\Theta\left(-\xi_{\mathbf{k}}^\text{F}\right)}{\frac{P^2}{2m_\text{F}} + \frac{(\mathbf{P}-\mathbf{k})^2}{2m_\text{B}} - \mu_{\text{CF}} - i\Omega + \varepsilon_0} \tag{A.12}$$

$$\approx \int \frac{d\mathbf{k}}{(2\pi)^2} \frac{\Theta\left(-\xi_{\mathbf{k}}^\text{F}\right)}{\varepsilon_0} = \frac{n_{\mu_\text{F}}^0}{\varepsilon_0}, \tag{A.13}$$

where $n_{\mu_\text{F}}^0 = m_\text{F} \mu_\text{F} \Theta(\mu_\text{F})/2\pi$, and, to obtain (A.13), we have neglected all terms in the denominator of Eq. (A.12) except for the large energy scale $\varepsilon_0$.

## B    3-body $T$-matrix in 2D vacuum within Born approximation

We briefly illustrate the evaluation of the 3-body $T$-matrix within the Born approximation in the 2D vacuum. We start by recalling the Skorniakov-Ter-Martirosian equation for the 3-body $T$-matrix $T_3(k,p;K)$ (see Fig. 19) describing the scattering between a BF dimer and a fermionic atom of initial three-momenta $K - k$ and $k$ and final three-momenta $K - p$ and $p$, respectively [128, 136]:

$$T_3(k,p;K) = -G_\text{B}^0(K-k-p) - \int \frac{d\mathbf{q}}{(2\pi)^2} \frac{dq_0}{(2\pi)} G_\text{B}^0(K-k-q) G_\text{F}^0(q) T_2(K-q) T_3(q,p;K), \tag{B.1}$$

where we have introduced the three-vector notation $q = (\mathbf{q}, q_0)$, where $q_0$ is a frequency, and $G_\text{B}^0$ and $G_\text{F}^0$ are the Bose and Fermi Green's function in vacuum

$$G_s^0(q) = \frac{1}{q_0 - \mathbf{q}^2/2m_s + i0^+}, \qquad s = \text{B}, \text{F}. \tag{B.2}$$

Within the Born approximation, $T_3$ is replaced with the first term on the right-hand side of Eq. (B.1). Moreover, for low energy scattering one can set $k \simeq p \simeq 0$ and $K \equiv \{\mathbf{0}, -\varepsilon_0\}$, (taking into account the binding energy $\varepsilon_0$ of the BF dimer). Recalling that $T_2$ is a dimer propagator except for the form factor $2\pi\varepsilon_0/m_r$ (see Eq. (20)) due to its composite nature, the same normalization factor is applied to $T_3$, which thus becomes independent of the two-body scattering length $a_{\text{BF}}$

$$\frac{2\pi\varepsilon_0}{m_r} T_3(0,0;\{\mathbf{0}, -\varepsilon_0\}) \simeq -\frac{2\pi\varepsilon_0}{m_r} G_\text{B}^0(\mathbf{0}, -\varepsilon_0) = \frac{2\pi}{m_r}. \tag{B.3}$$

## C    Convergence of numerical integrals

The convergence of the frequency integrals appearing in expressions (11), (12), (15) and (16) is guaranteed by the presence of the causality factors $e^{i\Omega 0^+}$ and $e^{i\omega 0^+}$ but in an extremely

slow fashion. In order to avoid this problem altogether, we add to and subtract from the integrand an analytically integrable auxiliary function with the same asymptotic behavior. By doing so, the integral turns out to be absolutely convergent and can be calculated numerically up to a high-frequency cutoff. Beyond the latter, the integration is performed analytically when computing the self-energies (11)-(12) and the contact parameter (62), whereas it is neglected in the case of momentum distributions (15)-(16). The formulas presented in the next subsections illustrate these two different cases. The remaining integral of the added auxiliary function can be instead integrated analytically thanks to the convergence factor.

### C.1 Self-energies

Starting from the self-energy for bosons (9), we recast it as follows

$$\Sigma_{\mathrm{BF}} = \int \frac{d\mathbf{P}}{(2\pi)^2} \int_{-\infty}^{\infty} \frac{d\Omega}{2\pi} \left[ T(\mathbf{P}, \Omega) G_{\mathrm{F}}^0(\mathbf{P} - \mathbf{k}, \Omega - \omega) - \Gamma_{\mathrm{SC}}(\mathbf{P}, \Omega) G_{\mathrm{F}}^{00}(\mathbf{P} - \mathbf{k}, \Omega - \omega) \right]$$
$$+ \int \frac{d\mathbf{P}}{(2\pi)^2} \int_{-\infty}^{\infty} \frac{d\Omega}{2\pi} \Gamma_{\mathrm{SC}}(\mathbf{P}, \Omega) G_{\mathrm{F}}^{00}(\mathbf{P} - \mathbf{k}, \Omega - \omega) e^{i\Omega 0^+}, \tag{C.1}$$

where $G_{\mathrm{F}}^{00}$ is a bare fermionic Green's function with $\mu_{\mathrm{F}}^0 = 0^-$. The second frequency integral is evaluated via contour integration closing on the left-hand side and contributing with a pole and a branch cut when they fall in this region. Instead, the first frequency integration is split into three regions: $[-\infty, -\Omega_c] \cup [-\Omega_c, \Omega_c] \cup [\Omega_c, \infty]$. For $|\Omega| \geq \Omega_c$ the integral is carried out analytically exploiting the subleading asymptotic expression of the integrand. The remaining frequency range is instead treated numerically.

Regarding integration in momentum space, the angular integral can be carried out analytically, since the dependence on the angle appears only in the bare Green's function. Finally, radial integrals are evaluated numerically up to a natural cut-off given by the position of the highest Fermi step in the integrand function. In particular, we find one step arising from the pole of the many-body $T$-matrix and up to two steps coming from the poles of the fermionic Green's function.

The calculations for the fermionic self-energy $\Sigma_{\mathrm{F}}$ follow the same approach as above provided $G_{\mathrm{F}}^{00}$ and $G_{\mathrm{F}}^0$ are replaced with $G_{\mathrm{B}}^0$. Since the bosonic bare Green's function does not have a pole, the final momentum integration is carried out simply over a single finite interval.

### C.2 Momentum distributions and densities

The momentum distributions Eqs. (15)-(16) are given by

$$n_{\mathrm{F}}(\mathbf{k}) = \int_{-\infty}^{\infty} \frac{d\omega}{2\pi} G_{\mathrm{F}}(\mathbf{k}, \omega) e^{i\omega 0^+} \simeq 2 \int_0^{\omega_c} \frac{d\omega}{2\pi} \mathrm{Re} \left[ G_{\mathrm{F}}(\mathbf{k}, \omega) - G_{\mathrm{F}}^0(\mathbf{k}, \omega) \right] + \Theta\left(-\xi_{\mathbf{k}}^{\mathrm{F}}\right), \tag{C.2}$$

$$n_{\mathrm{B}}(\mathbf{k}) = -\int_{-\infty}^{\infty} \frac{d\omega}{2\pi} G_{\mathrm{B}}(\mathbf{k}, \omega) e^{i\omega 0^+}$$
$$\simeq -2 \int_0^{\omega_c} \frac{d\omega}{2\pi} \mathrm{Re} \left[ G_{\mathrm{B}}(\mathbf{k}, \omega) - G_{\mathrm{B}}^{0'}(\mathbf{k}, \omega) \right] + \frac{1}{2} \left[ \frac{\xi_{\mathbf{k}}^{\mathrm{B}} + \Sigma_{11}}{\sqrt{\left(\xi_{\mathbf{k}}^{\mathrm{B}} + \Sigma_{11}\right)^2 - \Sigma_{12}^2}} - 1 \right], \tag{C.3}$$

where the bosonic Green's function in the Bogoliubov approximation is given by

$$G_{\mathrm{B}}^{0'}(\mathbf{k}, \omega) = \left( i\omega - \xi_{\mathbf{k}}^{\mathrm{B}} - \Sigma_{11} + \frac{\Sigma_{12}^2}{i\omega + \xi_{\mathbf{k}}^{\mathrm{B}} + \Sigma_{11}} \right)^{-1}. \tag{C.4}$$

The frequency cutoff $\omega_c$ is set to $50000E_F$ so to make the contribution of the integrands negligible for $\omega \geq \omega_c$, whereas in the range $100E_F \leq \omega \leq \omega_c$ the following asymptotic formulas are exploited for the self-energies appearing in the above Green's functions

$$\Sigma_F(\mathbf{k},\omega) \xrightarrow{\omega\to\infty} n_0 \Gamma(\mathbf{k},\omega) - \left(\frac{C_{BF}}{4m_r^2}\right) G_B^0(\mathbf{k},-\omega), \tag{C.5}$$

$$\Sigma_B(\mathbf{k},\omega) \xrightarrow{\omega\to\infty} \Sigma_{11} + n_{\mu_F} T(\mathbf{k},\omega) - \left(\frac{C_{BF}}{4m_r^2}\right) G_F^0(\mathbf{k},-\omega). \tag{C.6}$$

The last terms in Eq. (C.2) and in Eq. (C.3) originate from the frequency integration of the added and subtracted Green's functions, $G_F^0(\mathbf{k},\omega)$ and $G_B^{0'}(\mathbf{k},\omega)$ respectively.

One then arrives at the densities

$$n_F = \int_0^\infty \frac{dk}{2\pi} k\, n_F(k) = \int_0^{k_c} \frac{dk}{2\pi} k\, n_F(k) + \frac{\frac{C_{BF}}{4m_r}}{2\pi\left(\frac{k_c^2}{2m_r} - \mu_B - \mu_F\right)}, \tag{C.7}$$

$$\begin{aligned}
n_B &= n_0 + \int_0^\infty \frac{dk}{2\pi} k\, n_B(k) \\
&= n_0 + \int_0^{k_c} \frac{dk}{2\pi} k\, n_B(k) + \frac{\frac{C_{BF}}{4m_r}}{2\pi\left(\frac{k_c^2}{2m_r} - \mu_B - \mu_F\right)} + \frac{m_B \Sigma_{12}^2}{8\pi\left(\frac{k_c^2}{2m_B} - \mu_B\right)},
\end{aligned} \tag{C.8}$$

where we have used the asymptotic expressions (60)-(61).

## C.3 BF contact parameter $C_{BF}$

The BF pairing contribution $C_{BF}$ to the Tan's contact parameter is proportional to the trace of the $T$-matrix [126], which is recast as follows

$$\frac{C_{BF}}{4m_r^2} = \int \frac{d\mathbf{P}}{(2\pi)^2} \int_{-\infty}^\infty \frac{d\Omega}{2\pi} T(\mathbf{P},\Omega) e^{i\Omega 0^+} \tag{C.9}$$

$$= \int \frac{d\mathbf{P}}{(2\pi)^2} \int_{-\infty}^\infty \frac{d\Omega}{2\pi} [T(\mathbf{P},\Omega) - \Gamma_{SC}(\mathbf{P},\Omega)] + \int \frac{d\mathbf{P}}{(2\pi)^2} \int_{-\infty}^\infty \frac{d\Omega}{2\pi} \Gamma_{SC}(\mathbf{P},\Omega) e^{i\Omega 0^+} \tag{C.10}$$

$$= 2 \int \frac{d\mathbf{P}}{(2\pi)^2} \left\{ \mathrm{Re}\left[ \int_0^{\Omega_c} \frac{d\Omega}{2\pi} \left[ T(\mathbf{P},\Omega) - \Gamma_{SC}(\mathbf{P},\Omega) \right] + \frac{2\pi(n_0 + n_{\mu_F})}{m_r^2} \frac{\pi/2 - i(\ln\Omega_c - \ln\varepsilon_0)}{\pi^2/4 + \ln^2\Omega_c - \ln^2\varepsilon_0} \right] \right.$$
$$\left. + \frac{2\pi}{m_r} \left[ \varepsilon_0 \Theta\left(-\xi_{\mathbf{P}}^{CF}\right) + \Theta(-z_c) \int_{z_c}^0 \frac{dx}{\ln^2(x - z_c) - \ln^2\varepsilon_0 + \pi^2} \right] \right\}, \tag{C.11}$$

where $\Omega_c$ is the high-frequency cutoff, the term in the second line of Eq. (C.11) comes from the analytical integration of the asymptotic form of $T(\mathbf{P},\Omega) - \Gamma_{SC}(\mathbf{P},\Omega)$ in the range $[\Omega_c, +\infty]$ with the definition $n_{\mu_F} = 2m_F\mu_F\Theta(\mu_F)$, whereas in the last line $z_c = P^2/2M - \mu_B - \mu_F$ and $\xi_{\mathbf{P}}^{CF} = P^2/2M - \mu_{CF}$. The remaining integrals over $\Omega$, $x$ and $P$ are evaluated numerically.

# D  Analytic expressions in the strong-coupling limit

## D.1  Composite-fermion density and Hugenholtz-Pines condition

The integral in expression (33) can be computed analytically, yielding

$$\frac{n_{\mathrm{CF}}}{n_{\mathrm{F}}} = \Theta(y_+)\left[\frac{y_+}{2} - \sqrt{4\bar{\Delta}_0^2 + \left(\bar{\tilde{\mu}}_{\mathrm{CF}} - \bar{\mu}_{\mathrm{F}} + \frac{y_+}{2}\right)^2} + \sqrt{4\bar{\Delta}_0^2 + \left(\bar{\tilde{\mu}}_{\mathrm{CF}} - \bar{\mu}_{\mathrm{F}}\right)^2}\right]$$
$$+ \Theta(y_-)\left[\frac{y_-}{2} + \sqrt{4\bar{\Delta}_0^2 + \left(\bar{\tilde{\mu}}_{\mathrm{CF}} - \bar{\mu}_{\mathrm{F}} + \frac{y_-}{2}\right)^2} - \sqrt{4\bar{\Delta}_0^2 + \left(\bar{\tilde{\mu}}_{\mathrm{CF}} - \bar{\mu}_{\mathrm{F}}\right)^2}\right], \qquad \text{(D.1)}$$

with $y_\pm$ solutions of the equations

$$\bar{\tilde{\xi}}_y^{\mathrm{CF}} + \bar{\xi}_y^{\mathrm{F}} \pm \sqrt{(\bar{\tilde{\xi}}_y^{\mathrm{CF}} - \bar{\xi}_y^{\mathrm{F}})^2 + 4\bar{\Delta}_0^2} = 0. \qquad \text{(D.2)}$$

Here $\bar{\tilde{\xi}}_y^{\mathrm{CF}} = m_{\mathrm{F}} y/M - \bar{\tilde{\mu}}_{\mathrm{CF}}$ and $\bar{\xi}_y^{\mathrm{F}} = y - \bar{\mu}_{\mathrm{F}}$, and all quantities with a bar on top are in units of $E_{\mathrm{F}}$.

The Hugenholtz-Pines condition (47) can be evaluated analytically and, in terms of the quantity

$$I_\pm(y) = \ln\left[\sqrt{4\bar{\Delta}_0^2 + \left(\bar{\tilde{\mu}}_{\mathrm{CF}} - \bar{\mu}_{\mathrm{F}} + \frac{y}{2}\right)^2} \pm \left(\tilde{\mu}_{\mathrm{CF}} - \bar{\mu}_{\mathrm{F}} + \frac{y}{2}\right)\right], \qquad \text{(D.3)}$$

reads

$$\mu_{\mathrm{B}} = -4\varepsilon_0\Big\{\big[I_-(y_+) - I_-(0)\big]\Theta(y_+) + \big[I_+(y_-) - I_+(0)\big]\Theta(y_-)\Big\}. \qquad \text{(D.4)}$$

In the special case $x = 1$ and for $g \to \infty$, one has $\Delta_0 \to 0$ (as it can be seen in Fig. 15) and Eq. (D.4) can be simplified as

$$\mu_{\mathrm{B}} = -4\varepsilon_0 \ln\left[1 + \frac{\mu_{\mathrm{CF}}}{\mu_{\mathrm{B}} + \varepsilon_0}\right], \qquad \text{(D.5)}$$

where $\tilde{\mu}_{\mathrm{CF}} = \mu_{\mathrm{CF}}$ since $\Sigma_{\mathrm{CF}}^0 = 0$ for $x = 1$, and the definition $\mu_{\mathrm{CF}} = \mu_{\mathrm{F}} + \mu_{\mathrm{B}} + \varepsilon_0$ has been used.

By introducing $t \equiv (\mu_{\mathrm{B}} + \varepsilon_0)/\mu_{\mathrm{CF}}$, one has $\mu_{\mathrm{B}}/\varepsilon_0 = t\mu_{\mathrm{CF}}/\varepsilon_0 - 1$. To leading order in the small parameter $\mu_{\mathrm{CF}}/\varepsilon_0$, Eq. (D.5) reduces to $\frac{1}{4} = \ln\left[1 + \frac{1}{t}\right]$, with solution

$$t \equiv \frac{\mu_{\mathrm{B}} + \varepsilon_0}{\mu_{\mathrm{CF}}} = \frac{1}{e^{\frac{1}{4}} - 1}. \qquad \text{(D.6)}$$

For balanced populations, all bosons are expected to bind with fermions into pairs. This implies $n_0 = 0$ and, since $n_{\mathrm{CF}} = n_{\mathrm{F}}$, $\mu_{\mathrm{CF}} = E_{\mathrm{F}}/2$ (as one can verify using expression (33)). Equation (D.6), together with $\mu_{\mathrm{CF}} = E_{\mathrm{F}}/2$, analytically determine the values of both the fermionic and bosonic chemical potentials:

$$\mu_{\mathrm{B}} = -\varepsilon_0 + \frac{1}{e^{\frac{1}{4}} - 1}\frac{E_{\mathrm{F}}}{2}, \qquad \text{(D.7)}$$

$$\mu_{\mathrm{F}} = \frac{e^{\frac{1}{4}} - 2}{e^{\frac{1}{4}} - 1}\frac{E_{\mathrm{F}}}{2}. \qquad \text{(D.8)}$$

## D.2 Fermion momentum distribution for small hybridization energy $\Delta_0$

We now derive an analytic expression for the fermion momentum distribution in the strong-coupling limit when the hybridization energy $\Delta_0 \ll E_F$. Within our theoretical approach, this condition occurs either when $x \to 1$ (with the condensate density depleted by the formation of molecules) or when $x \to 0$ (with the condensate density small because of the small boson number). For intermediate concentrations, $\Delta_0$ remains instead of order $E_F$ in the strong-coupling limit (see Fig. 15). This analysis will provide us with a clear decomposition of $n_F(\mathbf{k})$ into its atomic and molecular contributions.

By expanding $G_F(\mathbf{k}, \omega)$ to first order in $\Delta_0/E_F$, one obtains

$$G_F(\mathbf{k}, \omega) \approx \frac{1}{i\omega - \xi_{\mathbf{k}}^F - \frac{\Delta_{CF}^2}{i\omega + \xi_{\mathbf{k}}^B}} + \frac{\Delta_0^2}{\left(i\omega - \tilde{\xi}_{\mathbf{k}}^{CF}\right)\left(i\omega - \xi_{\mathbf{k}}^F - \frac{2\pi n_{CF}}{m_r}\right)^2} , \tag{D.9}$$

where we have further approximated

$$\frac{\Delta_{CF}^2}{i\omega + \xi_{\mathbf{k}}^B} \approx -\frac{\Delta_{CF}^2}{\mu_B} = \frac{2\pi n_{CF}}{m_r} \equiv \Sigma_F^0 , \tag{D.10}$$

in the denominator of the second term. The quantity $2\pi n_{CF}/m_r \equiv \Sigma_F^0$ is analogous to $\Sigma_{CF}^0$ of Sec. 3.1 in that it represents the repulsion induced by the mean-field of the molecules onto the unpaired fermions and can be interpreted as a Hartree mean-field shift of the chemical potential ($\tilde{\mu}_F \equiv \mu_F - \Sigma_F^0$).

Let us first consider the first term in Eq. (D.9). It can be recast in the more familiar (BCS-like) form

$$\frac{\bar{u}_k^2}{i\omega - \bar{E}_{\mathbf{k}}^+} + \frac{\bar{v}_k^2}{i\omega - \bar{E}_{\mathbf{k}}^-} , \tag{D.11}$$

with

$$\bar{E}_{\mathbf{k}}^\pm = \frac{1}{2}\left(\xi_{\mathbf{k}}^F - \xi_{\mathbf{k}}^B \pm \sqrt{(\xi_{\mathbf{k}}^F + \xi_{\mathbf{k}}^B)^2 + 4\Delta_{CF}^2}\right) , \tag{D.12}$$

and

$$\bar{u}_k^2 = \frac{1}{2}\left(1 + \frac{\xi_{\mathbf{k}}^F + \xi_{\mathbf{k}}^B}{\sqrt{(\xi_{\mathbf{k}}^F + \xi_{\mathbf{k}}^B)^2 + 4\Delta_{CF}^2}}\right) , \qquad \bar{v}_k^2 = 1 - \bar{u}_k^2 . \tag{D.13}$$

We notice that in the asymptotic limit $\varepsilon_0 \gg E_F$, recalling that $\xi_{\mathbf{k}}^F + \xi_{\mathbf{k}}^B \approx \mathbf{k}^2/2m_r + \varepsilon_0$, we can expand the above quantities

$$\bar{E}_{\mathbf{k}}^+ \simeq \tilde{\xi}_{\mathbf{k}}^F , \qquad\qquad \bar{E}_{\mathbf{k}}^- \simeq -\xi_{\mathbf{k}}^B , \tag{D.14}$$

$$\bar{u}_k^2 \simeq 1 - \frac{\Delta_{CF}^2}{\left(\frac{\mathbf{k}^2}{2m_r} + \varepsilon_0\right)^2} , \qquad \bar{v}_k^2 \simeq \frac{\Delta_{CF}^2}{\left(\frac{\mathbf{k}^2}{2m_r} + \varepsilon_0\right)^2} , \tag{D.15}$$

having defined $\tilde{\xi}_{\mathbf{k}}^F \equiv \xi_{\mathbf{k}}^F + \Sigma_F^0$. Closing the integration contour on the left-hand side of the plane as prescribed by Eq. (15), from the first term in Eq. (D.9) one obtains the contribution

$$n_F^{(1)}(\mathbf{k}) = \left(1 - n_{CF}|\phi(\mathbf{k})|^2\right)\Theta\left(-\tilde{\xi}_{\mathbf{k}}^F\right) + n_{CF}|\phi(\mathbf{k})|^2 , \tag{D.16}$$

where

$$\phi(\mathbf{k}) = \sqrt{\frac{2\pi\varepsilon_0}{m_r}} \frac{1}{\frac{\mathbf{k}^2}{2m_r} + \varepsilon_0} , \tag{D.17}$$

is the bound-state wave-function for the two-body problem in vacuum, and $n_{CF}$ is determined by Eq. (33).

The second term in Eq. (D.16) is clearly associated with fermions belonging to molecules, as evidenced by the bound-state wavefunction $\phi(\mathbf{k})$. The first term describes instead unpaired fermions renormalized by the interaction, as evidenced by the presence of a Fermi step function with argument $-\tilde{\xi}_{\mathbf{k}}^{F}$ multiplied by a quasi-particle weight smaller than one.

We now deal with the second term in Eq. (D.9) which originates from the second diagram in Fig. 2(b). The pole in $z = \tilde{\xi}_{\mathbf{k}}^{CF}$ contributes

$$n_{F}^{(2)}(\mathbf{k}) = \frac{\Delta_0^2}{(\tilde{\xi}_{\mathbf{k}}^{CF} - \tilde{\xi}_{\mathbf{k}}^{F})^2}\Theta(-\tilde{\xi}_{\mathbf{k}}^{CF})\Theta(\tilde{\xi}_{\mathbf{k}}^{F}), \tag{D.18}$$

while the double pole in $z = \tilde{\xi}_{\mathbf{k}}^{F}$ contributes

$$n_{F}^{(3)}(\mathbf{k}) = -\frac{\Delta_0^2}{(\tilde{\xi}_{\mathbf{k}}^{CF} - \tilde{\xi}_{\mathbf{k}}^{F})^2}\Theta(\tilde{\xi}_{\mathbf{k}}^{CF})\Theta(-\tilde{\xi}_{\mathbf{k}}^{F}). \tag{D.19}$$

The fermion momentum distribution is thus obtained by adding the contributions of the three poles: $n_F(\mathbf{k}) = \sum_{i=1}^{3} n_F^{(i)}(\mathbf{k})$. More physically, it can be interpreted as the sum of a term $n_{CF}|\phi(\mathbf{k})|^2$ describing fermions bound into molecules, originating from the contribution (D.16) and a term $n_{UF}(\mathbf{k})$ corresponding to unpaired fermions:

$$n_F(\mathbf{k}) = n_{CF}|\phi(\mathbf{k})|^2 + n_{UF}(\mathbf{k}), \tag{D.20}$$

where

$$n_{UF}(\mathbf{k}) = \left[1 - n_{CF}|\phi(\mathbf{k})|^2 - \frac{\Delta_0^2\Theta(\tilde{\xi}_{\mathbf{k}}^{CF})}{(\tilde{\xi}_{\mathbf{k}}^{CF} - \tilde{\xi}_{\mathbf{k}}^{F})^2}\right]\Theta(-\tilde{\xi}_{\mathbf{k}}^{F}) + \frac{\Delta_0^2\Theta(-\tilde{\xi}_{\mathbf{k}}^{CF})}{(\tilde{\xi}_{\mathbf{k}}^{CF} - \tilde{\xi}_{\mathbf{k}}^{F})^2}\Theta(\tilde{\xi}_{\mathbf{k}}^{F}). \tag{D.21}$$

The above expression for $n_{UF}(\mathbf{k})$ can be simplified if either $\tilde{\mu}_{CF}$ or $\tilde{\mu}_F$ is negative such that the corresponding dispersion $\tilde{\xi}_{\mathbf{k}}^{CF}$ or $\tilde{\xi}_{\mathbf{k}}^{F}$ is always positive and the $\Theta$ functions become identically equal to zero or one. Within the present theory, the solution of the coupled Eqs. (45)−(47) yields indeed $\tilde{\mu}_F > 0$, $\tilde{\mu}_{CF} < 0$ for $x \to 0$ and $\tilde{\mu}_{CF} > 0$, $\tilde{\mu}_F < 0$ for $x \to 1$. One thus obtains

$$n_{UF}(\mathbf{k}) = \begin{cases} \left[1 - n_{CF}|\phi(\mathbf{k})|^2 - \dfrac{\Delta_0^2}{(\tilde{\xi}_{\mathbf{k}}^{CF} - \tilde{\xi}_{\mathbf{k}}^{F})^2}\right]\Theta(-\tilde{\xi}_{\mathbf{k}}^{F}), & x \to 0, \quad \text{(D.22a)} \\[4mm] \dfrac{\Delta_0^2}{(\tilde{\xi}_{\mathbf{k}}^{CF} - \tilde{\xi}_{\mathbf{k}}^{F})^2}\Theta(-\tilde{\xi}_{\mathbf{k}}^{CF}), & x \to 1. \quad \text{(D.22b)} \end{cases}$$

Expressions (D.22a) and (D.22b) describe unpaired fermions in two different regimes. In Eq. (D.22a) fermionic atoms are the majority species and most of them are unpaired because of the small number of bosons to pair with. When one inserts the contribution (D.22a) in Eq. (D.20) one recovers a nearly filled Fermi sphere for $k < \sqrt{2m_F\tilde{\mu}_F} \simeq k_F$, with a small depletion described by the last term in Eq. (D.22a). After the Fermi step, only the small contribution for fermions bound in molecules remains.

Equation (D.22b) describes instead unpaired fermionic atoms in a regime in which most fermions are bound in molecules. These few unpaired fermionic atoms partially occupy states with momentum $k < \sqrt{2m_F\tilde{\mu}_{CF}} \simeq k_F$ with a small probability $\Delta_0^2/(\tilde{\xi}_{\mathbf{k}}^{CF} - \tilde{\xi}_{\mathbf{k}}^{F})^2 \simeq \Delta_0^2/E_F^2 \ll 1$.



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
