# Peer review of "Boson-fermion pairing and condensation in two-dimensional Bose-Fermi mixtures"

_SciPost Physics, doi:SciPost Phys. 18, 076 (2025)_

## Round 1 · Referee Report · Anonymous (Referee 1) · 2024-6-25

Report
The manuscript titled "Boson-fermion pairing and condensation in two-dimensional Bose-Fermi mixtures" investigates the phase diagram of a two-dimensional mixture of a Bose gas and a Fermi gas which are coupled via a strong interaction that can feature bound states even in the two-body limit. The work builds on previous analyses conducted in three dimensions which utilized the same type of diagrams. Due to the presence of condensed bosons, the fermionic and the molecular sector are hybridized into a joint excitation and thus the relevant quasiparticles are superpositions of these two sectors. To illuminate this mechanism, the authors derive analytic expressions --valid in the strong-coupling regime-- which showcase the different quasiparticles at play. In the relevant regimes, their fully numerical treatment shows good overlap with their analytic expressions, and they find subtle physical differences (and also similarities) compared to the corresponding physical system in three dimensions. While, as in 3D, they find that increasing the Bose-Fermi interaction strength progressively depletes the condensate, they do not find the condensate fraction to vanish beyond a critical interaction strength (unlike in 3D). As a result, the emerging quasiparticle show some unusual features. Like in 3D, a universal behavior with respect to the ratio between bosonic and fermionic density is found for quantities like the condensate fraction.
The present manuscript is timely, and the physics covered is certainly interesting. I am not aware of previous works that have addressed this phase diagram along with the interplay of pairing and condensation in two dimensions. As mentioned in the main text, an experimental implementation of an analogous system in three dimensions was demonstrated recently and thus two-dimensional implementations are certainly within reach (and have already been studied in the polaron limits). Insights into the qualitative physics in two dimensions along with the differences to analogous systems in three dimensions are thus welcome and provide great progress to the field.
While this manuscript is certainly worthy of publication, there are a few issues/comments/questions I would like to see addressed before recommending publication. I provide the comments below.
Before getting into content feedback, I would like to encourage the authors to proofread the manuscript thoroughly. While reading through the manuscript, I found several instances of unnecessarily long/complicated sentences that required several readings before I was able to understand them. I also found several sentences missing verbs or with singular/plural errors. I invite the authors to conduct another round of proofreading to facilitate readability for all readers.
While the technical analysis conducted in this manuscript is sound (though in some instances I think the presentation may be improved), my main concerns are the diagrammatics used and the physical conclusions one may draw from it. I am fully aware, that the present analysis is challenging enough as is and going beyond it, taking into account higher-order effects or higher degrees of self-consistency is no easy feat. I feel however, that this manuscript lacks transparency in its main drawback, and I would like to see this discussed more. As I am sure the authors are fully aware, in the normal phase this type of T-matrix approach has a central drawback: it treats renormalization of the molecule sector and renormalization of the fermion sector on a different footing. As a result, in the Fermi-polaron limit of n_B=0, n_F>0 in three dimensions the molecule's energy is higher than it should be, and the polaron-to-molecule transition takes place later than it does in more self-consistent approaches. Apart from that, the Ansatz gives qualitatively correct results, especially in observables that do not include the energy (such as the quasiparticle weight). In two dimensions, however, this Ansatz no longer holds a polaron-to-molecule transition, even though several state-of-the-art methods do find one. This of course has implications also for the phase diagram and phase transitions of 2D B-F mixtures, as the present approach cannot reproduce known physics for x->0 at large g. This does not mean that at larger values of x it must produce incorrect results, but it is at least a possibility.
The authors mention this point in the conclusion, but I feel that in the interest of transparency it must be stated also in other places in the main text, especially when the diagrammatic is motivated due to its success in 3D or when results for large g and smaller values of x are discussed. It is also not clear to me how, when having a vanishing condensate fraction for x=0, this is regenerated upon going to x>0. Of course, I cannot discount the possibility that in 2D no phase transition exists, and one may rather see somewhat of a crossover, but this cannot be inferred from the analysis conducted here and I feel that the reader should be made aware of this potential shortfall BEFORE they get to the conclusion. I strongly encourage the authors to make mention of this, especially when pointing out differences between two and three dimensions, which may possibly only be due to this simple shortfall. I don't think mentioning it lessens the great value of their work.
In Figure 14, no correspondence between the condensate fraction and the Z factor is found. This is highly surprising to me as from a simple Chevy Ansatz one can see a close correspondence between the two. Is this difference because the chosen values of x were not small enough? If true, this is a major difference between 2D and 3D. Certainly one which has nothing to do with the absence of a polaron-molecule transition. Could the authors illuminate/investigate this further? For x->0 or at least for x=0, these two observables should be the same, so I am very surprised they are different.
Other feedback (in no particular order):
-
While I am familiar with the diagrammatics used, I am afraid many readers might not find them easily accessible. I feel that confusion might originate from the differences between the Gamma and T vertex in Figure 1 and how these diagrams are obtained. I suggest to either a) expand on an explanation of how the diagrammatics can be obtained (possibly in an appendix), b) provide a reference where this is done, or c) conduct parts of the explanation in a two-channel language (done recently in Ref. 53), where their different roles are more clear. I am not aware of Ref 73 or Ref 88 providing a more accessible explanation. In section 2B the Gamma vertex is referred to as a T-matrix; while I am aware that for n0=0 they are the same thing, this can easily add to confusion. I suggest to use distinct wording when referring to these vertices. Furthermore, I think a few references to T matrices in the normal phase might be helpful where the T matrix is first introduced.
-
I appreciate the analysis conducted in section 3 to illuminate the nature of the quasiparticles resulting from hybridization, however in part I find it very hard to follow due to its very technical nature and the non-trivial effects of hybridization. The analysis in 3A is easy to follow, I would possibly ask that the E^+ and E^- excitations are related also to the undressed states they correspond to for vanishing condensate density. The analysis in section 3B I found very hard to follow and I wonder whether it is necessary in the place it occurs now. It is my understanding that section 3 serves to illuminate the underlying quasiparticle qualitatively, mainly to shine light on the effects of hybridization:
In section 3A the T-matrix is analyzed, and one finds that it mixes unpaired atoms with molecules. The quantum fluctuations taken into account are a self-energy renormalization of the molecule (contained within Gamma). As a result, the two excitations found within T are a result of the mixing of a non-renormalized atom with a renormalized molecule. These two excitations would also show up in the same way if one considered a corresponding fermionic Green's function (of course the distribution of quasiparticle weight between them would be different). However, the fermionic Green's function in this work is considered on different footing and mixes a renormalized molecule with a renormalized fermion, resulting in two (slightly) different excitations. I am however not sure that this different, second set of excitations, introduced in 3B and analyzed after Eq. 32 for \Delta_0/E_F small is actually needed in this detail. Maybe the part with \Delta_0/E_F small is better suited for an appendix? When references to the analytical expressions were made later in the text, I could only really find references to 3A and parts up to Eq. 32.
The reason I bring this up, is that I found this part (after Eq. 32 until before Eq. 47) extremely challenging to follow, due to its technical nature and I am not sure which physical insights are conveyed in it. Furthermore, it had the effect that in section 3D and onwards I found it challenging to follow which analytical results were being referred to, those mainly from 3A or those from 3B.
-
Are there cheap ways to enforce a polaron-to-molecule transition, along with the corresponding physics? For example, in the analytical results obtained in Section 3. If yes, how do the results obtained from that look like?
-
Using the same diagrammatics in 3D, in Figure 7 of Ref. 77 a peculiar bosonic distribution function was observed, which vanished identically below a certain momentum. This was due to the bosons participating in fermionic molecule formation and as a result, the bosonic distribution function showed remnants of a fermionic effect. Can something analogous be observed here in 2D?
-
I find the used terminology of dressed/undressed dangerous. As is, "dressed" refers to the effects of hybridization (which one may see as resulting from a Green's matrix inversion in a two-channel language) and "undressed" refers to effects without hybridization. However, the undressed propagator still contains quantum fluctuations. In section 3A the adjective "bare" is used below equation 25. However, that molecule still contains quantum fluctuations, albeit in a mean-field fashion through Sigma_CF. Maybe the authors mean undressed? In any case, I feel these ambiguities can be a source of confusion for many readers and I would ask that the authors explain exactly what they mean when they use adjectives like dressed/undressed/bare.
-
I believe an analytical expression for Eq. 5 was provided in Ref. 106
-
Where is the condensate factor introduced? It briefly appears in the caption to Figure 1 and in the main text it starts appearing in Eq. 6, but there is no proper mention of what it actually is there.
-
In Eq. 8 the convergence factor appears but there is no mention of it.
-
In Eq. 2, it is v_0 while other times it is \nu_0.
-
I believe the renormalization/regularization in Eq.2 would benefit from a reference.
-
Is Omega a real or an imaginary frequency?
-
Are there things one can say about T>0?
-
The authors set m_B=m_F, what is the role of mass ratio in this phase diagram?
-
At the end of section 2A where the quantum depletion is mentioned, I think a reference to condensation in a repulsive Bose gas would be helpful, or alternatively an explanation of what is meant by "quantum depletion determined by eta_B"
-
Is the self-energy integral in Eq. 10 the same as the one introduced in Eq. 8? If yes, why not write Sigma_B=Sigma_11+ Sigma_BF?
-
"The direct boson-boson repulsion is neglected in the present regime, since it is expected to produce negligible effect", I am guessing this is for realistic experimental values?
-
I could not find any detail as to how Eq. 19 and 20 were obtained exactly
-
Is there a simple expression for n^0_{\mu F} below Eq. 20?
-
When G^0_CF is introduced, should the reader know already what it is or is it simply a Green's function of the form in Eq. 21 which is rescaled to fulfill the frequency sum rule of Green's functions?
-
Above Eq. 29 it is stated that the pole from G^0_B does not contribute. I am guessing this is due to the sign of the bosonic chemical potential. Was the sign of the bosonic potential already mentioned at this point? Does it ever change?
-
What does the sentence below Eq. 30 about neglecting E^\pm altogether with respect to mu_B mean?
-
Section 3B, I feel it could be made clearer where small Delta_0/E_F is presumed and where one goes back to considering Eq. 32.
-
"Before passing to the exact evaluation of Eq. (32) in closed form, it is instructive for its physical interpretation to analyze the limit of small Delta_0/F, which is expected to occur either when x -> 1 (depletion of the condensate density due to increase of molecule number) or x-> 0 (reduction of the condensate density due to decrease of boson number)." How can this be understood intuitively? I understand that for x->0 we have n_B->0 and since n_0<n_B we also have n_0->0. But how can this be intuitively understood for x->1? If I understand correctly, then for all n_B<n_F, all bosons could potentially bind into molecules, leaving n_0=0. Why is this only expected for x->1?
-
I believe Eqs. 41,43 and 44 were never formally related to n_F(k).
-
Why does the part in Eq. 45 that regards the population of composite fermions not have a corresponding theta function? Is there a way to understand Eqs. 45 and 46 in terms of two Fermi seas filling up? Or possible in terms of particle branches that are present in the two-body limit/ the non-interacting limit/ the Fermi polaron limit, which are then populated (albeit with modified quasiparticle properties)?
-
What is on the x-axis in the inset of Figure 3B? I presume x? I think Figure 3 would benefit from showing the condensate fraction.
-
In the paragraph before the one starting with "In the crossover region, all chemical potentials are comparable in size", how can one understand the characterizations of the particle in terms of their chemical potentials? Is there an intuitive way to understand why for example \tilde{\mu}_{CF}>0 means degenerate molecules.
-
I could not find it explicitly: Which method was used to obtain the results shown in Figure 3,4,5,6? Is it Eqs. 50-52 to obtain chemical potentials and condensate density or are additional Eqs. involved?
-
"In Fig. 5 the quasi-particle weight \nu^2_p of the occupied states \Theta(-E^-) is displayed for a number of concentrations corresponding to...". What does this mean? If I understand correctly then \nu^2_p*\Theta(-E^-) is shown. In Figure 5, the qp-weight is also shown for unoccupied states, however for unoccupied states it is set to 0. That is not precisely the same thing. Similarly, in the caption of Figure 5 only \nu^2_p is mentioned, but from Eq. 27 it is not clear to me that the function has a step-like drop (that only occurs after multiplying with Theta). Can this be formulated more precisely?
-
I think the paragraph below Figure 5 would benefit from explicitly stating what the Luttinger theorem is, how it breaks down, and how it is compensated in n_CF. I am guessing Theta has larger support and in return \nu^2_p is smaller, such that n_CF remains constant?
-
Eq. 61 is first referenced below Figure 8, a long time before it is first introduced.
-
I think more detail surrounding Eq.57 is needed. How can one see that a=1? What are the values of a, b as function of g? As is there is no real way to follow the argument made here.
-
"using the same thermodynamic parameters for both sets of curves". What does this mean? Same chemical potentials/condensate densities? If yes, how is this justified, shouldn't the different methods come up with different chemical potentials?
-
"The boundary between them can be roughly estimated by looking at the smallest coupling at which \mu_F vanishes as a function of x...". Is this a coincidence or is there physical meaning to \mu_F=0? I guess this goes back to my question of how to characterize particles in terms of their chemical potentials.
-
The notation inf Eqs 60,61 can be confusing, I would suggest switching to n_{F,k\gg k_F}(k) or something of that sort.
-
I found it hard to follow what happens in Appendix B. I think a Feynman diagram would help.
Recommendation
Ask for minor revision
Author: Pierbiagio Pieri on 2024-11-29 [id 5009]
(in reply to Report 1 on 2024-06-25)Please find our response to the Referee's report in the attached file
Author: Pierbiagio Pieri on 2024-11-29 [id 5008]
(in reply to Report 2 on 2024-07-05)Please find our response to the Referee's report in the attached file.
Attachment:
Referee2_2DBF.pdf

---

## Round 1 · Referee Report · Anonymous (Referee 2) · 2024-7-5

Report
The paper by L. Pisani presents a detailed study of imbalanced Bose-Fermi (BF) mixtures in cold atomic setups. It extends the T-matrix approach, which has been successful for the polaronic theories, to the arbitrary imbalance. The diverse range of observables is discussed in detail.
I have found that the calculations are solid, and the results are very interesting. The analysis of the observables presented is detailed and thoughtful. The paper will impact the field and be noticed by the cold atomic community. I will recommend its publication in the SciPost after the Authors clearly respond to the following points.
1) The Authors state that “However, since the latter choice gives rise to an improper self-energy contribution, to avoid double counting the -matrix in the normal phase Γ(P,Ω) is used instead, thus yielding…”. It is not clear what the improper means here. It looks like the leading contribution (according to the expansion of the Bogoliubov theory) has just been omitted. This approximation needs to be explained in detail.
2) The choice of diagrams is motivated by the well-established polaronic regime with the significant BF imbalance, i.e x<<1. It seems natural that it would produce reasonable results when x is modestly small. How the choice of diagram is justified at x~1? For instance, only particle-particle scattering diagrams are taken into account. What about the particle-hole ones? They do not produce bound states but can renormalize chemical potentials, broaden spectral weights, or modify masses.
2) The described physics has recently been found relevant and fruitful for electron-exciton systems in semiconductor nanostructures. The paper would benefit from discussing whether the obtained results can be extended to the mentioned solid-state setups.
Recommendation
Ask for minor revision

---

## Round 2 · Referee Report · Anonymous (Referee 2) · 2024-12-6

Report

The Authors have responded to all my points and made the corresponding changes in the paper.

The calculations are solid, and the results are very interesting. The analysis of the observables presented is detailed and thoughtful. The paper will impact the field and be noticed by the cold atomic community. I recommend its publication in SciPost in the present form.

Recommendation

Publish (easily meets expectations and criteria for this Journal; among top 50%)

---

## Round 2 · Referee Report · Anonymous (Referee 1) · 2025-1-13

Report

The authors have responded to my points in a satisfactory manner and they have made corresponding changes, that -in my eyes- greatly improve the quality of the manuscript as well as its accessibility.

The physics is interesting and with experimental realizations of such systems within reach this work is certainly timely. The present manuscript sets a foundation for the exploration of two-dimensional strongly-coupled Bose-Fermi mixtures, that future works will be able to draw from. As a result, I can recommend publication of the manuscript in its present from.

Below, I provide additional feedback that I encourage the authors to consider/incorporate if it is feasible and applicable. These points, I think, could enhance the insights obtained from this manuscript, but, if the authors deem these points not applicable or feasible, then my points should not delay publication. The manuscript in present form is already suitable for publication.

My feedback:

I still believe that one of the key results of this work, the condensate fraction differing from the quasiparticle weight, could be illuminated more.

The correspondence in 3D is fairly strong, which suggest that it isn’t a coincidence and also begs the question what changes in 2D.

I do not find the argument given by the authors on how „there is no reason why the limit for x->0 of the condensate fraction and the polaron residue Z should coincide“ particularly convincing. Both quantities are well defined in the thermodynamic limit. The field theory does not know about a particle number, it only knows about particle densities. The condensate fraction is obtained at nB>0, while the polaron quasiparticle weight is obtained from the weight of the quasiparticle pole in the bosonic Green’s/spectral function at nB=0 (which is also well defined at nB>0). Importantly the quasiparticle weight obtained from the spectral function has the same physical interpretation as the quasiparticle weight in the polaron wave function (Chevy) Ansatz.

In both cases one has taken V->infinity before specifying the chemical potentials (and condensate densities) which eventually yield the corresponding densities nB and nF.

Furthermore, it was my understanding that the universality came from, as the authors note also in this work, the bosons being nearly independent of each other, while interacting with the medium. Thus, it would at least be physically intuitive that the probabilities for a boson to be in a p=0 mode are related in these cases. My understanding was that the correspondence between the quasiparticle weight and the condensate fraction was a reflection of that. So if a universality is observed here, then I would still expect some sort of correspondence between probabilities to be in effect. Thus it would be insightful to illuminate where the „remaining probability“ goes.

-The curves shown in Figure 14 are for eta=0, however for eta=0 we have that nB(0) is finite. For eta\neq 0, nB(0) diverges. Could it be that there is some sort of delta function for p=0 that contributes to the fraction of p=0 bosons? Perhaps because in 2D one only has a single factor of p in the measure instead of p^2 in 3D? Do plots like Figure 14a also exist for eta>0? Could it be that there is stronger correspondence for eta\neq 0?

-How does the bosonic quasiparticle weight at x>0 (not just at x=0) compare to the condensate fraction?

-I may be mistaken about this: Have the authors considered reconstructing the quantum effective action from the renormalizations employed here? In particular the effective potential? After a short, (not very careful) analysis I obtain that \frac{\delta \Gamma[\phi[J]]}{\delta \phi[J]} |_{J=0}= \phi[0] G^{-1}_B[\phi[0]]. Where J is the source field, \phi[J] is the source-dependent boson field and \phi[0] is the stationary field at vanishing source, which here is proportional to \phi[0]\propto \sqrt{rho} \delta(…).
It would seem that the Hugenholtz-Pines condition employed here is a necessary condition for the field to be stationary. However, I don’t think it necessarily implies that \phi[0] minimizes the effective potential? This could either mean that the value obtained for rho is not unique and there is a second value of rho that fulfills the Hugenholtz-Pines condition, along with the other fixing conditions AND additionally yields a lower value of the effective potential. Alternatively, it could also mean that the field additionally condenses in a different mode, for example p>0 (though I don’t think this is the case here).
Have the authors considered checking if there is a larger value of \rho that fulfills the Hugenholtz-Pines condition? I would guess that actually computing the effective potential and comparing values is quite cumbersome, but checking whether there is a second solution to the Hugenholtz-Pines condition should be feasible.

-I believe a formal consideration of the effective potential/ effective action might also yield more insights into the possible correspondence between quasiparticle weight and condensate fraction

-I may be mistaken, but is there a possibility that the used Hugenholtz-Pines condition is only a low-order approximation of a „more“ accurate condition? Eq. 4.10 in Ref. 107 and the text below seem to indicate that Eq. 6.2 of Ref. 107 is only an approximation. Though it seems to be increasingly valid at low density, which however begs the question of whether this would refer to boson or fermion density in this case? I am not sure how this point fits with my previous points, but I thought it might be better to include it regardless.

Recommendation

Publish (surpasses expectations and criteria for this Journal; among top 10%)

  • validity: -
  • significance: -
  • originality: -
  • clarity: -
  • formatting: -
  • grammar: -

Author:  Leonardo Pisani  on 2025-01-29  [id 5160]

(in reply to Report 2 on 2025-01-13)
Category:
reply to objection

Please find our response to the Referee's report in the attached file.

Attachment:

Response_II_round_BF2D.pdf

---

## Round 2 · Author Response

Dear Editor,

Thank you for your Editorial Recommendation of July 5th, 2024, asking for a minor revision of our manuscript. We hereby submit a revised version that takes into due account all recommendations by the Referees. Together with this resubmission, we have also sent a point-by-point response to both Referees.

We are really grateful to the Referees for their valuable comments and suggestions, which have helped us to considerably improve our manuscript.

Yours sincerely,
Leonardo Pisani, Pietro Bovini, Fabrizio Pavan, Pierbiagio Pieri

---

## Round 2 · List of Changes

Summary of the changes made

We have changed the layout of our text to adopt the SciPost latex template. We refer to the new version when pointing to page, equation, and figure numbers. Our manuscript has been modified to take into account all the suggestions by the Referees. We have also made some changes to improve the presentation and quoted a few additional references. All of these changes (except minor stylistic/typos corrections) are listed below.

  1. Pag. 2, changed “micro-cavities” with “nanostructures” in the 2nd paragraph, added four new sentences to the 2rd paragraph (“The so-called … on more solid ground.”)

  2. Pag. 3, re-written the final part of the 5th paragraph (from “For this reason …”)

  3. Pag. 5, added a sentence at the end of the first paragraph, re-worded the first two sentences of the last paragraph of Sec. 2.1, re-worded the end of the first paragraph in Sec. 2.2 (“There, the authors … to the boson concentration.”), added a new paragraph at the end of the page (starting with “A warning is however in order.”)

  4. Pag. 6, added two new paragraphs (“Let us see now the rationale … controlled approximations are not available”, “The same strategy is then adopted … with (one-component) fermions [76].”), inserted Eq.5 and corresponding discussion

  5. Pag. 7, re-worded the paragraph above Eq.7 and the paragraph above Eq.9, added a comment below Eq.9, introduced more specific names for the T-matrices (“many-body” and ”two-body”) below Eq.9

  6. Pag. 8, re-written Eq.11 and the paragraph below it

  7. Pag. 9, changed the titles of Sec.3 and Sec.3.1, re-worded the last two sentences of the first paragraph of Sec.3 and the first sentence of Sec. 3.1, added the explicit definition of n^0_{mu_F} below Eq.21, changed Sigma_{CF} to Sigma^0_{CF} below Eq.22 and everywhere else

  8. Pag. 10, added Eq.24 and its explanation below it, added the sentence “in the many-body T-matrix in the condensed phase” above Eq.25, added a new paragraph (“Here, G_{CF}(P,Omega), … condensate.”) below Eq.27, re-worded the first sentence below Eq.29, added a whole new paragraph (“One sees from Eq. (29) … value for all intermediate concentrations.”) above Eq.30, changed “pair” with “composite-fermion” before Eq.30

  9. Pag. 11, added Eq.34 and comments above and below it, changed “two-body correlator” with “many-body T-matrix” at the end of Sec.3.1, re-worded the paragraph containing Eqs. 35 and 36

  10. Pag. 12, re-organized the end of Sec.3.2, in particular moving text and equations from former Sec.IIIB (from former Eq.33 to 46b) to the new Appendix D2 while re-organizing and re-wording the rest; re-worded the sentence after Eq.44

  11. Pag. 13, added a comment at the end of the paragraph after Eq.47, re-written the last two paragraphs of the page

  12. Pag. 14, re-written the 1st paragraph, added a second inset in Fig.3(b) updating the caption, re-worded the last sentence of the page

  13. Pag. 15, added a sentence “So, in this regime, …”, modifies the first sentence of the 2nd paragraph, re-written the last two sentences of the 2nd paragraph, inserted new paragraph “The Luttinger theorem states …” at the end of the page

  14. Pag. 16, inserted new paragraph (2nd paragraph: “In a BF mixture with a condensate …”), updated the caption of Fig.5, completely re-written the last paragraph of the page with several new sentences

  15. Pag. 17, completely re-written all paragraphs before Section 4 with several new sentences, added a new sentence (last sentence of the page)

  16. Pag. 19, added Eq.49

  17. Pag. 20, added insets in Fig.9 and their explanations in the caption

  18. Pag. 21, re-worded a sentence in the caption of Fig.11, added two new sentences after Eq. 51

  19. Pag. 22, substituted “asymptotic limit” with “large-momentum behavior” in the caption of Fig.12, added two new sentences in the 2nd paragraph: “to check to what extent …” and “The use of the same thermodynamic…”,, added “internal part of the” before “molecular wave-function” in the first sentence of the 3rd paragraph

  20. Pag. 23, replaced “asymptotics” with “approximation” in the caption of Fig.13, added a new sentence immediately after Eq.55

  21. Pag. 24, added a new curve and a new set of points in Fig.14(a) updating the corresponding caption, re-written a sentence in the 1st paragraph, inserted a new paragraph (2nd paragraph), re-worded the last sentence of Sec.4.3

  22. Pag. 25, added two new sentences (“As an independent check…to possible numerical errors”) near the end of the 3rd paragraph, added a new 4th paragraph (“As a matter of fact, … followed by the limit x -> 0.”), added four new sentences at the end of the last paragraph

  23. Pag. 26, changed the notation of Eqs.56-57

  24. Pag. 27, reworded the last sentence of the last paragraph at the end of the page

  25. Pag. 28, re-written the first two sentences of the 4th paragraph in Sec.5 “A further difference with …”

z) Pag. 29, added words “in particular at small values of x …” at the end of the second sentence of the 2nd paragraph, added a new paragraph at the end of the page (”It is worth also discussing the relevance … sufficient to guarantee stability.“) aa) Pag. 30, added two new paragraphs (“As a final remark …are available online [131].”) before the Acknowledgements bb) Pag. 30-31, renamed, re-organized, and re-written Appendix A cc) Pag. 32, added Fig.17 and its caption dd) Pag. 34-35, added Appendix D1 containing the whole content of former Appendix D ee) Pag. 35-37, added Appendix D2 containing material from former Sec.IIIB (from former Eq.33 to 46b) ff) Bibliography, added the following references 42, 44, 48, 49, 95, 96, 100, 101, 102, 111, 112, 113, 114, 115, 128, 129, 130, 131 gg) Bibliography, updated the following references (due to publication or change of version): former 34 now 39, former 45 now 46, former 53 now 56 hh) Bibliography, removed former reference 40

---

## Round 3 · Author Response

Dear Editor,

Thank you for your Editorial Recommendation of Jan 13th, 2025.
As you recommended, we have addressed the point that Referee 2 suggested to further consider before publication. We hereby submit a revised version that takes into due account this point. Together with this resubmission, we have also sent a point-by-point response to the last report by Referee 2.

We hope that with this last improvement our manuscript can proceed to publication on SciPost Physics. We thank again the Referees for their valuable comments and suggestions.

Yours sincerely,

Leonardo Pisani, Pietro Bovini, Fabrizio Pavan, Pierbiagio Pieri

---

## Round 3 · List of Changes

a) Sec. 4.3, inserted new figure (Fig. 14), with caption
b) Sec. 4.4, added new panel (c) to Fig. 15 (previous Fig. 14) and modified the caption accordingly
c) Added new paragraph "In this respect, reconsidering the 3D case ..." (4th paragraph of Sec. 4.4)
d) Sec. 4.4, added Eq. (56).
e) Sec. 4.4 added paragraph “An interesting related question is …” after Eq. 56, containing new Eqs. 57,58,59
f) Sec. 4.4, added paragraph “To further analyze this behavior at more generic BF coupling …” after Eq. (59)
g) Sec. 4.4, inserted new figure (Fig. 16), with caption
h) Corrected a typo in Eq. A.9
i) Corrected a typo in Eq. D.10
j) Bibliography, added references 108, 109, 124, 125
k) Bibliography, updated Refs. 38 and 56 to published versions

---

## Editorial Decision

published